# Clinical efficacy of different therapeutic options for knee osteoarthritis: A network meta-analysis based on randomized clinical trials

Xiao Chen[1]◉, Yuanhe Fan[1]◉, Hongliang Tu[1]◉, Yuan Luo◉[2]*

**1** Department of Orthopedics, the First People's Hospital of Neijiang, Neijiang, China, **2** Department of Rehabilitation, the First People's Hospital of Neijiang, Neijiang, China

◉ These authors have contributed equally to this work.
* 15583216261@163.com

## Abstract

### Objective

To assess and compare the clinical efficacy of various therapeutic options in treating patients with knee osteoarthritis (KOA).

### Methods

We performed a comprehensive search of PubMed, Embase, OVID, Cochrane Library and Web of Science databases from their inception to December 10th, 2023, identifying randomized controlled trials (RCTs) examining the effects of therapeutic options on KOA. Two researchers independently performed literature screening, data extraction, data collection and organization, and quality assessment. The data obtained were subjected to statistical analysis and graphical representation using Stata 17.0 software.

### Results

A total of 139 RCTs encompassing 9644 KOA patients and involving 12 therapeutic options were included. These interventions were low level laser therapy (LLLT), high intensity laser therapy (HILT), transcutaneous electrical nerve stimulation (TENS), interferential current (IFC), short wave diathermy, ultrasound, lateral wedged insole, knee brace, exercise, hydrotherapy, kinesio taping (KT) and extracorporeal shock wave therapy (ESWT). Regarding the WOMAC pain score, knee brace was determined to be the most likely to yield the best results, followed by exercise and HILT, ultrasound was worst intervention. In terms of WOMAC function score, knee brace emerged as the technique with the highest likelihood of being optimal, followed in sequence by hydrotherapy and ESWT, ultrasound was worst intervention. Knee brace ranked highest in effectiveness concerning the WOMAC stiffness score, followed by

**Data availability statement:** All relevant data are within the manuscript and its Supporting Information files.

**Funding:** The author(s) received no specific funding for this work.

**Competing interests:** The authors have declared that no competing interests exist.

exercise and hydrotherapy. For the total WOMAC score, hydrotherapy demonstrated the highest probability of being the best technique, followed by exercise and HILT, short wave diathermy was worst intervention. In addressing VAS-rest, hydrotherapy exhibited the greatest likelihood of being the optimum technique, followed by HILT and LLLT. In terms of VAS-activity, knee brace had the highest probability of being the best technique, followed by LLLT and exercise, ultrasound was worst intervention. Overall, based on the results obtained from the SUCRA for all outcomes, knee brace had the highest probability of being the best technique, followed by hydrotherapy and exercise.

## Conclusion

The findings suggest that knee brace may be the most recommended therapeutic option for the knee osteoarthritis, followed by hydrotherapy and exercise.

## 1 Introduction

Knee osteoarthritis, a common chronic condition in the middle-aged and elderly, is clinically characterized by the degenerative changes within the knee joint. It presents with pain, stiffness, and functional limitations related to knee inflammation and effusion, potentially adversely impacting the quality of life [1]. Statistics indicate that more than 10% of individuals over the age of 60 are affected by this disease, which, along with its associated pain and functional disability, as well as the societal costs of joint replacement surgeries, represents a considerable burden [2].

Knee osteoarthritis (KOA) treatment strategies include a spectrum of options such as pharmacotherapy, interventional treatments, regenerative therapies involving cellular and acellular approaches, and joint replacement surgeries [3]. Nonsteroidal Anti-Inflammatory Drugs (NSAIDs) and acetaminophen are the most frequently prescribed analgesics for managing pain and enhancing physical function in osteoarthritis patients. Estimates indicate that between 10% to 35% of osteoarthritis patients utilize oral NSAIDs or acetaminophen for symptom management. However, the consumption of these medications is linked to gastrointestinal and cardiovascular adverse events and potentially an elevated risk of mortality, particularly among elderly patients with concurrent health issues [4,5].

Physical therapy offers a safer alternative to pharmacological interventions with a lower incidence of side effects. It is essential to investigate the efficacy of various physical therapy methods in the treatment of KOA. The array of physical therapy techniques includes low level laser therapy, high intensity laser therapy, transcutaneous electrical nerve stimulation, interferential current, short wave diathermy, ultrasound, lateral wedged insole, knee brace, exercise, hydrotherapy, kinesio taping and extracorporeal shock wave therapy. While existing research has primarily focused on comparing physical therapy with other treatment options, there is a paucity of studies that directly assess the comparative efficacy of different physical therapy approaches. This gap in the literature limits our ability to discern the relative advantages and disadvantages of each method in the context of KOA patient care [6].

Consequently, the objective of this study is to conduct a network meta-analysis to evaluate the clinical efficacy of the aforementioned physical therapy interventions for KOA. The findings aim to equip clinical practice with the valuable evidence-based medical evidence.

## 2 Method

### 2.1 Protocol and registration

This study adheres to the Preferred Reporting Items for Systematic Reviews and Meta-Analyses (PRISMA) 2020 statement, ensuring a structured methodology and reporting format, and A MeaSurement Tool to Assess systematic Reviews (AMSTAR) 2 guidelines. Furthermore, the network meta-analysis protocol has been duly registered on PROSPERO data base (the registration number: CRD42023473080).

### 2.2 Data sources

A comprehensive literature search was conducted by two independent researchers, with any disparities resolved through consultation with a third investigator. The search encompassed titles and abstracts, and full-text assessments were carried out as needed to determine study eligibility.

The following databases were systematically searched from their inception until December 10th, 2023: PubMed, Embase, OVID, Cochrane Library, Web of Science, and Scopus databases. Eligible studies encompassed only randomized controlled trials involving participants diagnosed with knee osteoarthritis. These studies compared various therapeutic options. The search utilized the following relevant terms: "osteoarthritis", "knee", "laser", "transcutaneous electrical nerve stimulation", "interferential current", "short wave", "ultrasound", "wedged insole", "brace", "exercise", "hydrotherapy", "kinesio taping", "extracorporeal shock wave therapy" and "random". Additionally, Google Scholar was searched to identify potentially relevant literature. Furthermore, the reference lists of identified reports were meticulously reviewed to identify any additional pertinent studies. Only articles published in the English language were considered for inclusion in our network meta-analysis.

### 2.3 Eligibility criteria

The inclusion criteria were as follows: (1) Participants: Aged 18 years and above, diagnosed with unilateral or bilateral KOA based on the American College of Rheumatology criteria, categorized as mild-to-moderate KOA according to the Kellgren-Lawrence radiographic classification, and presented with knee pain. If both knees were affected, the knee with worse symptoms was included in the outcome assessment; (2) Types of Studies: Relevant randomized controlled trials (RCTs); (3) Types of Interventions: Studies comparing any combination of the 12 therapeutic options mentioned above and placebo were included. Each study needed to involve at least two of these therapies. (4)Types of Outcomes: Western Ontario and McMaster Universities Osteoarthritis Index (WOMAC) (pain, stiffness, function and total score) and visual analog scale (VAS) (rest and activity) at last follow-up.

The exclusion criteria were as follows: (1) non-RCTs, Case reports, review articles, meta-analyses, editorials, letters, animal studies, and cadaveric trials. (2) Patients with a previous history of knee surgery, decompensated organ failure, treatment for oncological diseases, systemic infammatory diseases, infectious diseases, intra-articular injection within the last 6 months. (3) Physical therapy or balneological treatment within the last 1 year, and change in the drug routine within last 2 months. (4) Studies with incomplete or missing data. (5) Duplicate articles. (6) Poor-quality research literature or studies lacking rigor in their design. (7) Papers with abstracts only and RCT protocols. (8) Non-English articles.

### 2.4 Data extraction

A specifically designed form was employed to extract essential information from each enrolled study. The data that was extracted included: (1) General information such as the lead author, year of publication, study design, country of study,

study period, and follow-up time; (2) Demographic information, including the number and proportion of male or female patients, age at diagnosis, and the number of patients involved; (3) Details regarding the therapeutic options (intervention and comparison); (4) Information on clinical outcomes, including the pain, stiffness, function and total score of WOMAC and VAS (rest and activity) at last follow-up. In instances where SD was not available from the publication, SD was imputed using the method prescribed in the Cochrane Handbook.

## 2.5 Quality assessment

For RCTs, the Cochrane Risk of Bias Tool was employed to assess the quality. The risk of bias for the included trials was evaluated by two researchers (the first and second authors) based on the Cochrane Handbook criteria.

## 2.6 Statistical analysis

To conduct a comprehensive network meta-analysis, we utilized the statistical software packages "Network" and "mvmeta" within STATA 17.0 software. Continuous variables including WOMAC and VAS were analyzed using weighted mean differences (WMD) with corresponding 95% CI. When the 95% CI of the contained the value 1, the comparison was considered statistically non-significant.

For direct comparisons, a conventional meta-analysis was conducted to aggregate the results using random-effects models, serving as sensitivity analyses. The network meta-analysis employed a frequentist approach with a random-effects model to estimate both direct and indirect comparisons. The primary objective of the network meta-analysis was to assess whether any of the comparator interventions demonstrated superiority. To evaluate potential inconsistencies between indirect and direct comparisons, we employed global inconsistency, local inconsistency (using a node-splitting approach), and loop inconsistency. Statistical significance for global inconsistency was determined using P-values, with $P > 0.05$ indicating no significant global inconsistency. Local inconsistency was assessed through node-splitting analysis, and $P > 0.05$ indicated no significant local inconsistency. Heterogeneity within each closed loop was estimated using the inconsistency factor (IF), with a 95% CI (IF) value of zero signifying no statistical significance. In each pre-specified outcome, a global network diagram was employed to illustrate direct comparisons between interventions. The size of the nodes in the diagram corresponded to the number of participants receiving each treatment. Treatments subject to direct comparisons were linked by lines, and the thickness of these lines was proportional to the number of trials evaluating the specific comparison.

All intervention measures were ranked based on their respective SUCRA values or the area under the curve, resulting in a comprehensive ranking of the interventions. To assess the potential for publication bias, the comparison-adjusted funnel plot was utilized. This analysis aimed to determine whether there was evidence of a small sample effect or publication bias within the intervention network.

## 3 Results

### 3.1 Search results

A total of 3424 studies were initially identified from various sources, including PubMed (n = 2252), Embase (n = 229), Ovid (n = 465), Web of Science (n = 418), and the Cochrane Library (n = 60). After removing duplicate and irrelevant literatures, a full text search was conducted for the remaining 295 literatures. Ultimately, 139 studies, involving 9644 patients, met the eligibility criteria for inclusion in this network meta-analysis. Participants: This study included 9644 patients with KOA (age ≥ 18 years). And it involved 12 interventions: LLLT, HILT, TENS, IFC, short wave diathermy, ultrasound, lateral wedged insole, knee brace, exercise, hydrotherapy, KT, and ESWT. The control was a placebo. Outcomes included WOMAC pain, function, and stiffness scores at last follow-up, WOMAC total score, and VAS scores at rest and after activity. The process of study selection is illustrated in **Fig 1**, and the baseline characteristics of the included studies are summarized in **Table 1**.

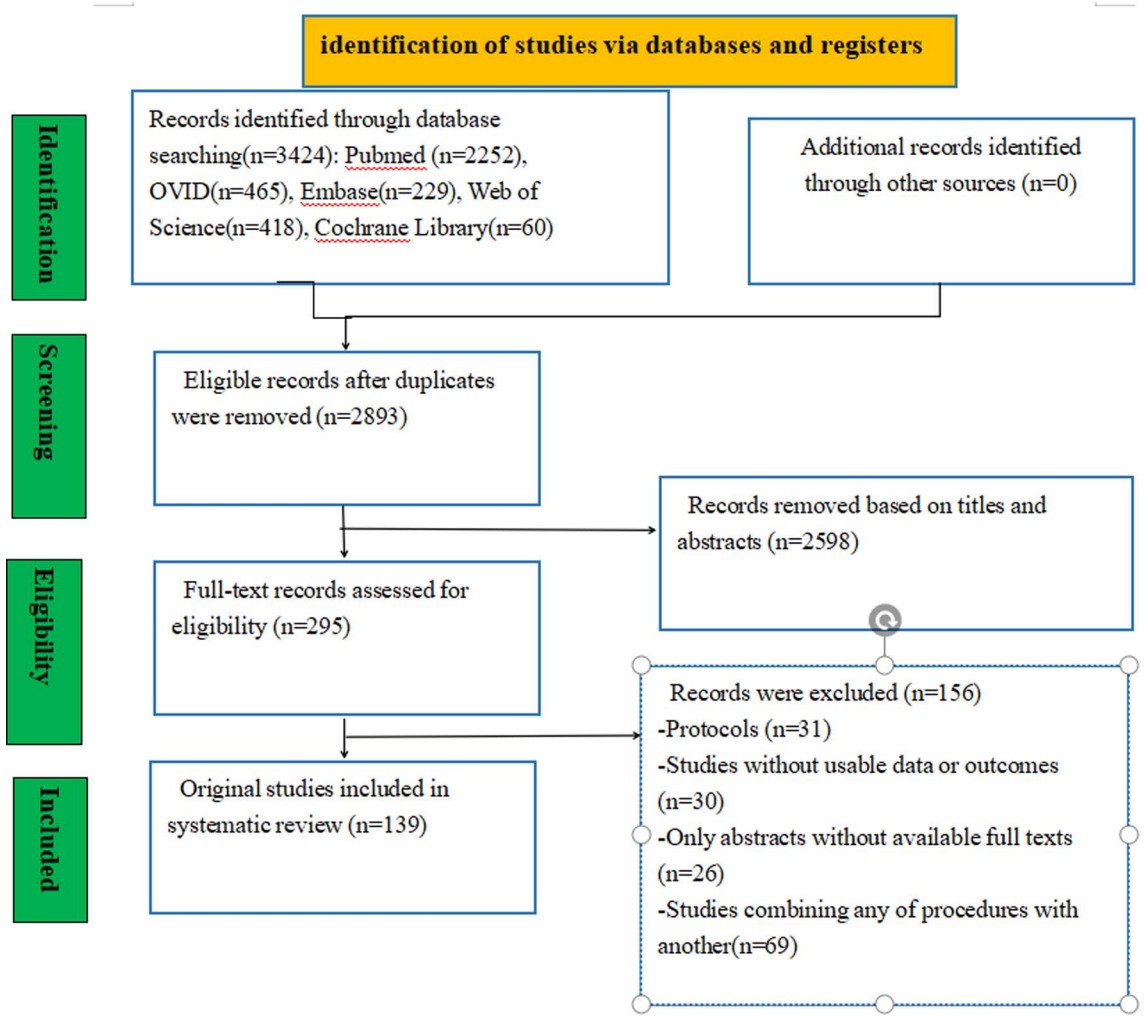

**Fig 1. Flow chart of included studies.**

## 3.2 Risk of bias and quality assessment

The quality assessment of the included 139 RCTs was conducted using the Cochrane Collaboration's "Risk of Bias" tool. The risk of bias assessment for the included studies is presented in **Table 2**.

## 3.3 Evidence network

This study encompassed 12 distinct physical therapies, including LLLT, HILT, TENS, IFC, short wave diathermy, ultrasound, lateral wedged insole, knee brace, exercise, hydrotherapy, KT and ESWT. **Fig 2** visually represents the evidence network, where the lines denote direct comparisons between two directly related interventions. Interventions lacking direct connections are compared indirectly through the network meta-analysis. The width of the lines reflects the number of trials, while the size of the nodes corresponds to the total sample size across multiple treatments.

**Table 1. Baseline characteristics of the included studies.**

| – | Country | Study design | Intervention | N | Sex (M/F) | Age (years) | BMI | Follow-up | Outcome |
|---|---|---|---|---|---|---|---|---|---|
| Adıgüzel 2021 | Turkey | RCT | Hydrotherapy | 32 | 1/31 | 60.84±7.76 | 30.34±4.33 | 24 weeks | WOMAC, VAS |
| | | | Placebo | 32 | 1/31 | 59.09±9.15 | 31.25±5.15 | | |
| Ahmad 2023 | Malaysia | RCT | HILT | 17 | 3/14 | 51.18±9.79 | 30.58±5.43 | 12 weeks | VAS |
| | | | LLLT | 17 | 5/12 | 57.94±10.56 | 27.57±4.47 | | |
| Ahmad 2023 | Malaysia | RCT | Exercise | 40 | 19/21 | 65.27±6.86 | 28.83±3.38 | 8 weeks | VAS |
| | | | Placebo | 40 | 15/25 | 65.60±8.70 | 26.86±4.37 | | |
| Akaltun 2021 | Turkey | RCT | HILT | 20 | – | 57.85±8.06 | 29.93±5.49 | 8 weeks | WOMAC, VAS |
| | | | Placebo | 20 | – | 58.61±11.28 | 31.95±4.86 | | |
| Akyol 2010 | Turkey | RCT | Short wave diathermy | 20 | 0/20 | 57.8±10.65 | 31.05±5.22 | 12 weeks | WOMAC, VAS |
| | | | Placebo | 20 | 0/20 | 56.6±8.13 | 30.37±3.25 | | |
| Alfredo 2020 | Brazil | RCT | Ultrasound | 20 | 6/14 | 64.35±6.16 | 30.98±3.36 | 8 weeks | WOMAC, VAS |
| | | | Placebo | 20 | 6/14 | 62.7±8.49) | 31.09±3.16 | | |
| Alfredo 2022 | Brazil | RCT | LLLT | 20 | 4/16 | 68.55±9.62 | 28.39±4.35 | 24 weeks | WOMAC |
| | | | Placebo | 20 | 3/17 | 65.9±8.82 | 29.16±3.65 | | |
| Alghadir 2013 | Saudi Arabia | RCT | LLLT | 20 | 10/10 | 55.2±8.14 | 32.34±5.77 | 4 weeks | WOMAC, VAS |
| | | | Placebo | 20 | 12/8 | 57±7.77 | 33.09±4.98 | | |
| Ali 2014 | Pakistan | RCT | TENS | 25 | – | 50.5±4.6 | – | 4 weeks | WOMAC |
| | | | Placebo | 25 | – | 50.5±4.6 | – | | |
| Alireza 2023 | Iran | RCT | Exercise | 24 | 4/20 | 52.8±9.6 | 29.22±3.77 | 8 weeks | WOMAC, VAS |
| | | | Placebo | 24 | 7/17 | 55.7±9.2 | 31.3±5.03 | | |
| Alkhawajah 2019 | Arabia | RCT | Exercise | 20 | 13/7 | 56.5±7.6 | 32.6±7.8 | 2 days | WOMAC |
| | | | Placebo | 20 | 12/8 | 56.6±8.5 | 33.3±6.1 | | |
| Allen 2021 | UA | RCT | Exercise | 230 | 194/36 | 59.9±9.9 | 33.9±7.4 | 36 weeks | WOMAC |
| | | | Placebo | 115 | 98/17 | 60.2±11.1 | 33.9±7.5 | | |
| Rashoud 2014 | UK | RCT | LLLT | 26 | 10/16 | 52±9 | 38±5.6 | 24 weeks | VAS |
| | | | Placebo | 23 | 8/15 | 56±11 | 37.1±5.3 | | |
| Alqualo-Costa 2021 | Brazil | RCT | IFC | 42 | 13/29 | 64.5±7.8 | 29.3±5.2 | 24 weeks | WOMAC |
| | | | Placebo | 42 | 14/28 | 65.3±8.5 | 29.9±4.6 | | |
| Amornthep 2023 | Taiwan | RCT | LLLT | 16 | 1/15 | 67.44±6.54 | – | 1 week | WOMAC |
| | | | Placebo | 16 | 5/11 | 71.63±7.60 | – | | |
| Anandkumar 2014 | India | RCT | KT | 20 | 8/12 | 55.9±5.0 | – | – | VAS |
| | | | Placebo | 20 | 9/11 | 55.7±5.8 | – | | |
| Anwer 2014 | India | RCT | Exercise | 21 | – | 54.9±7.7 | 26.5±1.8 | 6 weeks | WOMAC |
| | | | Placebo | 21 | – | 56.0±6.8 | 27.1±1.3 | | |
| Artuc 2023 | Turkey | RCT | TENS | 19 | 3/16 | 58.50±9.80 | 30.33±6.62 | 12 weeks | WOMAC, VAS |
| | | | IFC | 18 | 4/14 | 61.95±11.78 | 31.43±3.51 | | |
| Assar 2020 | Iran | RCT | Exercise | 12 | – | 57.5±6.9 | 28.5±3.7 | 8 weeks | WOMAC, VAS |
| | | | Placebo | 12 | – | 63.8±7.5 | 23.1±11.6 | | |
| Atamaz 2012 | Turkey | RCT | TENS | 37 | 6/31 | 61.9±6.9 | 28.4±3.5 | 24 weeks | WOMAC, VAS |
| | | | IFCs | 31 | 4/21 | 62.0±7.9 | 29.8±3.4 | | |
| | | | Short wave diathermy | 31 | 4/27 | 61.6±7.4 | 28.5±4.2 | | |
| Aydogdu 2017 | Turkey | RCT | KT | 28 | – | 52.53±9.68 | 31.18±5.14 | 12 weeks | WOMAC |
| | | | Placebo | 26 | – | 51.19±8.94 | 31.52±5.70 | | |
| Baykal 2023 | Turkey | RCT | KT | 82 | 18/64 | 66.04±6.36 | – | – | VAS |
| | | | Placebo | 82 | 20/62 | 64.98±5.78 | – | | |

*(Continued)*

**Table 1.** (Continued)

| – | Country | Study design | Intervention | N | Sex (M/F) | Age (years) | BMI | Follow-up | Outcome |
|---|---|---|---|---|---|---|---|---|---|
| Bennell 2015 | Australia | RCT | Exercise | 73 | 23/50 | 67.4±8.6 | 29.3±4.3 | 48 weeks | WOMAC, VAS |
| | | | Placebo | 67 | 23/44 | 69.8±7.5 | 28.9±3.9 | | |
| Bennell 2011 | Australia | RCT | Lateral wedge insole | 103 | 41/62 | 63.3±8.1 | 28.1±4.2 | 52 weeks | WOMAC, VAS |
| | | | Placebo | 97 | 31/56 | 65.0±7.9 | 30.4±5.6± | | |
| Bruce-Brand 2012 | Ireland | RCT | TENS | 10 | 4/6 | 63.9±5.8 | 33.7±5.6 | 6 weeks | WOMAC |
| | | | Exercise | 10 | 4/6 | 63.4±5.9 | 33.9±8.3 | | |
| | | | Placebo | 6 | 3/3 | 65.2±3.1 | 31.7±4.1 | | |
| Cantista 2020 | Spain | RCT | Hydrotherapy | 60 | – | – | – | 12 weeks | WOMAC, VAS |
| | | | Placebo | 60 | – | – | – | | |
| Carpenedo 2021 | Italy | RCT | Ultrasound | 8 | 2/6 | 70.37±7.36 | 29.48±4.42 | 24 weeks | VAS |
| | | | Placebo | 8 | 3/5 | 70.87±11.81 | 29.62±3.43 | | |
| Chen 2014 | China | RCT | Exercise | 30 | – | >40 | – | 24 weeks | WOMAC, VAS |
| | | | Ultrasound | 30 | – | >40 | – | | |
| | | | ESWT | 30 | – | >40 | – | | |
| | | | Placebo | 30 | – | >40 | – | | |
| Chen 2023 | China | RCT | HILT | 158 | 35/123 | 64.74±6.38 | 24.98±3.68 | 12 weeks | WOMAC |
| | | | Placebo | 151 | 42/109 | 63.93±5.86 | 24.64±9.15 | | |
| Chen HX 2021 | China | RCT | Exercise | 10 | 3/7 | 59.63±8.40 | 23.24±0.85 | 6 weeks | WOMAC, VAS |
| | | | Placebo | 8 | 2/6 | 58.30±8.54 | 23.02±1.37 | | |
| Chen PY 2021 | China | RCT | Exercise | 36 | 4/32 | 77.4±5.9 | 24.7±2.6 | 12 weeks | WOMAC, VAS |
| | | | Placebo | 32 | 2/30 | 75.4±6.4 | 24.0±2.7 | | |
| Cherian 2016 | USA | RCT | TENS | 33 | 14/19 | 58 (33–77) | – | 48 weeks | VAS |
| | | | Placebo | 37 | 10/27 | 62 (27–86) | – | | |
| Cho 2015 | South Korea | RCT | KT | 23 | 6/17 | 58.2±4.5 | – | – | VAS |
| | | | Placebo | 23 | 7/16 | 57.5±4.4 | – | | |
| Choi 2023 | Korea | RCT | ESWT | 9 | 4/5 | 73.7±2.4 | 25.6±2.9 | 4 weeks | WOMAC, VAS |
| | | | Placebo | 9 | 5/4 | 72.6±2.3 | 26.4±2.0 | | |
| Danazumi 2021 | Nigeria | RCT | KT | 30 | – | 52.3±5.19 | 24.1±4.08 | – | VAS |
| | | | Placebo | 30 | – | 52.0±6.25 | 23.9±4.23 | | |
| Dantas 2023 | Brazil | RCT | Exercise | 16 | 7/9 | 60.9±9.5 | 29.1±3.5 | 8 weeks | WOMAC |
| | | | Placebo | 14 | 8/6 | 63.1±8.2 | 30.8±3.9 | | |
| Dias 2016 | Brazil. | RCT | Hydrotherapy | 33 | 0/33 | 70.8±5.00 | 30.5±4.30 | 6 weeks | WOMAC, VAS |
| | | | Placebo | 32 | 0/32 | 71.0±5.20 | 30.0±5.20 | | |
| Dogan 2022 | Turkey | RCT | KT | 27 | 0/27 | 56.9±6.9 | 32.8±5.8 | 8 weeks | VAS |
| | | | Placebo | 30 | 0/30 | 55.7±6.9 | 30.8±5.4 | | |
| Donec 2020 | Lithuanian | RCT | KT | 81 | 17/64 | 68.7±9.9 | 30.5±5.3 | 4 weeks | VAS |
| | | | Placebo | 76 | 16/60 | 70.6±8.3 | 30.7±5.2 | | |
| Ekici 2022 | Turkey | RCT | HILT | 30 | 13/17 | 61.07±6.96 | 30.97±3.31 | 12 weeks | WOMAC, VAS |
| | | | Placebo | 26 | 7/19 | 57.85±7.04 | 32.17±4.79 | | |
| Elboim-Gabyzon 2023 | Israel | RCT | TENS | 20 | 7/13 | 62.7±6.6 | 31.1±3.3 | 3 weeks | WOMAC |
| | | | LLLT | 20 | 5/15 | 63.0±6.2 | 30.8±3.7 | | |
| Fazli 2023 | Iran | RCT | KT | 28 | 17/11 | 55.7±5.3 | 24.4±3.04 | 4 weeks | WOMAC, VAS |
| | | | Placebo | 28 | 11/17 | 54.4±3.2 | 23.2±2.7 | | |
| Fokmare 2023 | India | RCT | Hydrotherapy | 30 | 7/23 | 50.06±6.66 | – | 2weeks | WOMAC, VAS |
| | | | Knee brace | 30 | 9/21 | 51.43±4.88 | – | | |

*(Continued)*

| – | Country | Study design | Intervention | N | Sex (M/F) | Age (years) | BMI | Follow-up | Outcome |
|---|---|---|---|---|---|---|---|---|---|
| Foley 2014 | Australia | RCT | Hydrotherapy | 35 | 20/15 | 73.0±8.2 | – | 6 weeks | WOMAC |
| | | | Exercise | 35 | 15/20 | 69.8±9.2 | – | | |
| | | | Placebo | 35 | 15/20 | 69.8±9.0 | – | | |
| Foroughi 2011 | Australia | RCT | Exercise | 26 | 0/26 | 66±8 | 31.4±5.4 | 24 weeks | WOMAC |
| | | | Placebo | 28 | 0/28 | 65±7 | 32.7±8.4 | | |
| Fukuda 2011 | USA | RCT | Short wave diathermy | 30 | – | 62.0±8.0 | 29.4±4.5 | 48 weeks | WOMAC, VAS |
| | | | Placebo | 21 | – | 57.0±9.0 | 27.6±3.7 | | |
| Fukuda 2011 | USA | RCT | LLLT | 25 | 5/20 | 63.0±9.0 | 30.0±3.5 | 3 weeks | VAS |
| | | | Placebo | 22 | 8/14 | 63.0±8.0 | 28.7±4.1 | | |
| Fung 2021 | China | RCT | Exercise | 37 | 3/34 | 75.1±8.0 | 22.0±2.0 | – | WOMAC, VAS |
| | | | Placebo | 36 | 2/34 | 76.1±7.5 | 22.5±2.2 | | |
| Gao 2023 | China | RCT | Exercise | 13 | 5/8 | 68.54±2.07 | 26.38±1.99 | 8 weeks | VAS |
| | | | Placebo | 14 | 6/8 | 67.86±1.41 | 26.47±2.80 | | |
| Gholami 2023 | UK | RCT | Exercise | 32 | – | – | – | 24 weeks | WOMAC, VAS |
| | | | Placebo | 32 | – | – | – | | |
| Gomes 2020 | Brazil. | RCT | Placebo | 20 | 2/18 | 69.4±4.45 | – | 10 weeks | WOMAC |
| | | | TFC | 20 | 2/18 | 71.85±2.62 | – | | |
| | | | Short wave diathermy | 20 | 1/19 | 68.45±4.62 | – | | |
| | | | LLLT | 20 | 0/20 | 65.75±4.48 | – | | |
| Günaydin 2022 | Turkey | RCT | KT | 22 | 0/22 | 58.8±6.2 | 28.8±4.7 | 12 weeks | VAS |
| | | | ESWT | 18 | 0/18 | 58.8±6.2 | 28.8±4.7 | | |
| | | | Placebo | 20 | 0/20 | 58.8±6.2 | 28.8±4.7 | | |
| Guo 2021 | China | RCT | Placebo | 52 | 24/28 | 61.5±7.2 | 27.6±5.4 | 8 weeks | VAS |
| | | | Exercise | 50 | 23/27 | 63.2±8.0 | 27.4±6.0 | | |
| Gur 2003 | Turkey | RCT | LLLT | 30 | 7/23 | 59.80±8.03 | 28.49±3.02 | 14 weeks | WOMAC, VAS |
| | | | Placebo | 30 | 6/24 | 60.52±6.91 | 30.27±311 | | |
| Hammam 2020 | Egypt | RCT | EWST | 15 | 6/9 | 50.4±3.4 | 30.7±3.5 | 4 weeks | VAS |
| | | | Placebo | 15 | 8/7 | 49.7±3.1 | 31.1±3 | | |
| Han 2021 | China | RCT | Ultrasound | 31 | 13/18 | 53.6±19.1 | – | 24 weeks | WOMAC, VAS |
| | | | Placebo | 31 | 14/17 | 55.6±17.6 | – | | |
| Hinman 2003 | Australia | RCT | KT | 29 | 9/19 | 66±8 | 29.3±4.0 | 3 weeks | WOMAC, VAS |
| | | | Placebo | 29 | 9/19 | 69±9 | 30.1±4.0 | | |
| Ho 2022 | Taiwan | RCT | ESWT | 18 | 6/12 | 65.6±11 | 24.1±2.4 | 3 weeks | WOMAC, VAS |
| | | | Placebo | 18 | 5/13 | 64.6±11.8 | 23.9±1.4 | | |
| Hu 2019 | China | RCT | Exercise | 52 | – | 66.32±4.16 | 36.49±8.99 | 24 weeks | WOMAC, VAS |
| | | | Placebo | 40 | – | 65.54±3.59 | 26.4±3.07 | | |
| Iijima 2020 | Japan | RCT | TENS | 30 | 7/23 | 59.9±6.41 | 22.1±2.94 | – | VAS |
| | | | Placebo | 30 | 10/20 | 58.2±5.63 | 23.4±4.03 | | |
| Imamura 2016 | Brazil | RCT | ESWT | 52 | 0/52 | 70.0±6.5 | – | 12 weeks | WOMAC, VAS |
| | | | Placebo | 53 | 0/53 | 72.4±6.5) | – | | |
| Itoh 2008 | Japan | RCT | Placebo | 6 | – | 62-83 | – | 10 weeks | WOMAC, VAS |
| | | | TENS | 6 | – | 62-83 | – | | |
| Jang 2023 | Korea | RCT | Short wave diathermy | 53 | 1/52 | 61.45±5.05 | – | 4 weeks | WOMAC, VAS |
| | | | Ultrasound | 52 | 4/48 | 60.85±5.11 | – | | |

*(Continued)*

| – | Country | Study design | Intervention | N | Sex (M/F) | Age (years) | BMI | Follow-up | Outcome |
|---|---|---|---|---|---|---|---|---|---|
| Jia 2022 | China | RCT | Ultrasound | 57 | 15/42 | 62.28±10.88 | 25.18±3.26 | 24 weeks | WOMAC |
| | | | Short wave diathermy | 57 | 12/45 | 59.93±8.97 | 25.29±2.85 | | |
| Jones 2013 | UK | RCT | Lateral wedged insole | 28 | – | 66.3±8.2 | – | 2 weeks | WOMAC, VAS |
| | | | Knee brace | 28 | – | 66.3±8.2 | – | | |
| Jorge 2023 | Brazil | RCT | LLLT | 44 | 18/26 | 59.1±9.3 | 29.0±3.4 | 24 weeks | WOMAC, VAS |
| | | | Placebo | 42 | 16/26 | 58.4±8.3 | 29.9±3.4 | | |
| Karakas 2020 | Turkey | RCT | Ultrasound | 39 | 8/31 | 59.10±7.45 | 28.70±4.86 | 12 weeks | WOMAC, VAS |
| | | | Placebo | 36 | 4/32 | 60.75±7.46 | 29.22±10.13 | | |
| Karimi 2021 | Iran | RCT | Exercise | 10 | 0/10 | 40.61±8.54 | – | 8 weeks | WOMAC |
| | | | Placebo | 10 | 0/10 | 65.6±6.54 | – | | |
| Kayamutlu 2016 | Turkey | RCT | KT | 20 | 4/16 | 54.25±6.01 | 30.72±3.80 | 4 weeks | WOMAC, VAS |
| | | | Placebo | 19 | 2/17 | 57.10±6.26 | 31.34±6.16 | | |
| Kayamutlu 2018 | Turkey | RCT | Exercise | 21 | 5/16 | 56.29±6.64 | 30.74±4.31 | 48 weeks | VAS |
| | | | TENS | 22 | 3/19 | 57.77±6.24 | 32.59±5.70 | | |
| Kheshie 2014 | UK | RCT | HILT | 20 | – | 52.1±6.47 | 29.94±3.36 | 6 weeks | WOMAC, VAS |
| | | | LLLT | 18 | – | 56.56±7.86 | 28.62±5.2 | | |
| | | | Placebo | 15 | – | 55.6±11.02 | 28.51±3.35 | | |
| Khosravi 2021 | Iran | RCT | Knee brace | 7 | 3/4 | 59.2±8.07 | – | 6 weeks | WOMAC, VAS |
| | | | Lateral wedge insole | 7 | 4/3 | 60.3±5.28 | – | | |
| Khruakhorn 2021 | Thailand | RCT | Placebo | 17 | 1/16 | 57.88±7.75 | 27.27±4.38 | 24 weeks | WOMAC |
| | | | Hydrotherapy | 17 | 2/15 | 64.88±7.44 | 26.34±2.7 | | |
| Kilic 2020 | Turkey | RCT | Exercise | 25 | – | 59.52±8.57 | 30.58±2.94 | 6 weeks | WOMAC, VAS |
| | | | Placebo | 25 | – | 60.48±7.43 | 31.19±3.33 | | |
| Kitano 2023 | Japan | RCT | Ultrasound | 13 | 3/10 | 63.5±8.6 | 26.5±4.3 | 10 weeks | WOMAC, VAS |
| | | | Placebo | 13 | 2/11 | 56.5±7.5 | 24.0±5.2 | | |
| Kocyigit 2015 | Turkey | RCT | KT | 21 | 2/19 | 52±7.5 | – | 2 weeks | VAS |
| | | | Placebo | 20 | 3/17 | 52±10 | – | | |
| Laufer 2005 | Israel | RCT | Short wave diathermy | 38 | 7/31 | 74.79+6.58 | – | 12 weeks | WOMAC, VAS |
| | | | Placebo | 33 | 11/22 | 73.33+6.91 | – | | |
| Lee 2023 | Korea | RCT | Exercise | 15 | – | 65.63+3.70 | – | 8 weeks | VAS |
| | | | Placebo | 16 | – | 68.27+4.78 | – | | |
| Leon 2017 | Mexico | RCT | KT | 16 | – | 56.5±5.0 | 29.5±4.1 | 6 weeks | WOMAC, VAS |
| | | | Placebo | 16 | – | 59.6±5.2 | 29.4±3.2 | | |
| Lewinson 2016 | Canada | RCT | Lateral wedge insole | 19 | 6/13 | 59.9±7.4 | 32.5±8.0 | 12 weeks | VAS |
| | | | Placebo | 19 | 8/11 | 59.6±7.7 | 29.2±6.7 | | |
| Liao 2020 | Taiwan | RCT | LLLT | 15 | 1/14 | 70.53±6.89 | 26.61±4.33 | 4 weeks | VAS |
| | | | Placebo | 15 | 2/13 | 69.73±6.91 | 25.98±2.71 | | |
| Lin 2022 | Taiwan | RCT | Exercise | 20 | 1/19 | 75.6±4.4 | 24.6±3.5 | 12 weeks | WOMAC |
| | | | Placebo | 18 | 2/16 | 76.0±5.6 | 24.2±2.2 | | |
| Maheu 2022 | France | RCT | TENS | 55 | 18/37 | 66.9±8.1 | 28.0±5.0 | 12 weeks | WOMAC |
| | | | Placebo | 55 | 21/34 | 66.0±7.8 | 29.8±5.8 | | |
| Mahler 2018 | France | RCT | LLLT | 27 | 12/15 | 62±9 | 29(25–30) | 12 weeks | VAS |
| | | | Placebo | 28 | 15/13 | 68±9 | 26(24–31) | | |

*(Continued)*

| – | Country | Study design | Intervention | N | Sex (M/F) | Age (years) | BMI | Follow-up | Outcome |
|---|---|---|---|---|---|---|---|---|---|
| Marconcin 2021 | Portugal | RCT | Exercise<br>Placebo | 32<br>35 | 14/18<br>7/28 | 67.8±5.3<br>70.3±6.1 | 30.1±5.3<br>32.3±5.0 | – | VAS |
| Mascarin 2012 | Brazil | RCT | Exercise<br>TENS<br>Ultrasound | 16<br>12<br>12 | –<br>–<br>– | 59.6±7.2<br>64.8±7.0<br>62.8±7.6 | –<br>–<br>– | 12 weeks | WOMAC, VAS |
| McManus 2021 | Australia | RCT | Placebo<br>KT | 17<br>19 | 5/12<br>9/10 | 70±8.0<br>68±8.8 | 30.60±5.0<br>30.11±5.5 | 5 weeks | VAS |
| Messier 2022 | USA | RCT | Exercise<br>Placebo | 414<br>409 | 94/320<br>92/317 | 64.5±7.8<br>64.7±7.8 | 36.7±6.5<br>36.9±7.2 | 72 weeks | WOMAC |
| Messier 2022 | USA | RCT | Exercise<br>Placebo | 94<br>90 | 30/64<br>24/66 | 67.4±6.1<br>65.0±5.6 | 33.1±3.3<br>33.8±3.6 | 240 weeks | WOMAC |
| Mete 2022 | Turkey | RCT | Exercise<br>Placebo | 30<br>30 | 6/24<br>7/23 | 59.5(55-64)<br>57(51-65) | 30.3 (24.96-33.16)<br>32.14 (31.8-32.42) | 6 weeks | VAS |
| Mobina 2019 | Iran | RCT | Knee brace<br>Lateral edge insole | 7<br>7 | 3/4<br>4/3 | 59.2±8.07<br>60.3±5.28 | –<br>– | 6 weeks | WOMAC, VAS |
| Mohamed 2022 | Saudi Arabia | RCT | KT<br>Placebo | 20<br>20 | 20/0<br>20/0 | 60.60±9.43<br>63.40±7.98 | 26.65±2.85<br>27.72±2.41 | 6 weeks | WOMAC |
| MohammedSadiq 2021 | Iraq | RCT | Exercise<br>Placebo | 16<br>15 | 3/13<br>1/14 | 51.38±7.72<br>53.8±8.46 | 31.61±4.12<br>34.14±4.45 | 8 weeks | WOMAC |
| Mostafa 2021 | Egypt | RCT | ESWT<br>HILT | 20<br>20 | 9/11<br>10/10 | 40.12±9.45<br>46.62±8.68 | 28.82±5.23<br>29.26±2.48 | 4 weeks | WOMAC, VAS |
| Müller-Rath 2011 | US | RCT | Knee brace<br>Placebo | 13<br>10 | –<br>– | 30–60<br>30–60 | < 30<br>< 30 | 16 weeks | WOMAC, VAS |
| Nambi 2016 | Saudi Arabia | RCT | LLLT<br>Placebo | 17<br>17 | –<br>– | 58±6<br>60±8 | 26.9±4.8<br>28.3±3.5 | 8 weeks | VAS |
| Nazari 2018 | Iran | RCT | HILT<br>Exercise | 30<br>30 | 13/17<br>14/16 | 61.5±3.9<br>62.24±3.87 | 27.7±1.4<br>27.5±1.8 | 12 weeks | WOMAC |
| Oğuz 2021 | Turkey | RCT | KT<br>Placebo | 11<br>11 | 0/11<br>0/22 | 48.18±7.56<br>51.00±3.69 | 30.90±3.17<br>34.76±5.91 | 6 weeks | WOMAC, VAS |
| Palmer 2014 | UK | RCT | TENS<br>Placebo | 73<br>74 | 26/47<br>25/49 | 61.2±11.4<br>60.9±10.8 | 29.7±11.1<br>29.1±9.0 | 24 weeks | WOMAC |
| Park 2021 | South Korea | RCT | Exercise<br>Placebo | 25<br>25 | –<br>– | 66.88±4.61<br>68.04±4.16 | –<br>– | 8 weeks | WOMAC |
| Pierosimone 2020 | US | RCT | TENS<br>Placebo | 32<br>29 | 14/18<br>10/19 | 60.8±7.3<br>62.5±7.7 | 29.2±3.3<br>27.8±4.4 | 8 weeks | WOMAC |
| Pinto 2020 | Brazil | RCT | LLLT<br>Placebo | 15<br>16 | 1/14<br>1/15 | 63±2.83<br>66±2.69 | 24.8±2.4<br>29.8±1.1 | 5 weeks | VAS |
| Pozsgai 2022 | Hungary | RCT | Exercise<br>Placebo | 21<br>20 | 7/14<br>2/18 | 68.18±5.21<br>66.64±4.26 | 28.62±5.39<br>32.20±5.73 | 6 days | WOMAC |
| Qiestad 2023 | Norway | RCT | Exercise<br>Placebo | 53<br>54 | 25/28<br>30/24 | 57.3±7.1<br>57.8±7.4 | 29.4±4.4<br>28.4±4.1 | 48 weeks | WOMAC |
| Rabiei 2023 | Iran | RCT | Exercise<br>Placebo | 27<br>27 | 9/18<br>13/14 | 60.5±5.6<br>59.8±5.1 | 29.5±4.4<br>29.3±3.4 | 8 weeks | WOMAC |
| Rafiq 2021 | Malaysia | RCT | Exercise<br>Placebo | 28<br>28 | 11/17<br>9/19 | 54.21±5.20<br>55.00±4.86 | 33.91±5.66<br>31.03±2.78 | 12 weeks | WOMAC |
| Rahlf 2017 | Germany | RCT | KT<br>Placebo | 47<br>47 | 23/24<br>23/24 | 64.7±7.3<br>65.3±6.0 | –<br>– | 3 days | WOMAC |
| Rego 2023 | Brazil | RCT | Exercise<br>Placebo | 8<br>9 | 0/8<br>0/9 | 65.75±2.76<br>64.78±2.17 | 28.90±1.41<br>32.02±2.40 | – | VAS |

*(Continued)*

**Table 1.** (Continued)

| – | Country | Study design | Intervention | N | Sex (M/F) | Age (years) | BMI | Follow-up | Outcome |
|---|---|---|---|---|---|---|---|---|---|
| Reichenbach 2021 | Switzerland | RCT | TENS<br>Placebo | 108<br>112 | 56/52<br>52/60 | 64.8±9.9<br>66.3±10.3 | 27.5±4.9<br>26.9±4.9 | 3 weeks | WOMAC |
| Rewald 2019 | Netherlands | RCT | Exercise<br>Placebo | 47<br>55 | 23/24<br>16/39 | 61±7.4<br>59±9.5 | 29±5.4<br>29±5.6 | 6 weeks | WOMAC |
| Ridvan 2020 | Turkey | RCT | Short wave diathermy<br>Placebo | 31<br>32 | 14/17<br>13/19 | 62.78±7.53<br>58.68±8.15 | –<br>– | 12 weeks | WOMAC |
| Robbins 2021 | Australia | RCT | LLLT<br>Placebo | 43<br>43 | 11/32<br>5/38 | 66.09±5.89<br>62.44±3.34 | 32.89±5.95<br>31.69±3.79 | 11 weeks | WOMAC |
| Samaan 2022 | Egypt | RCT | HILT<br>Ultrasound<br>Placebo | 20<br>20<br>20 | 9/11<br>7/13<br>10/10 | 55.4±6.34<br>55.2±4.77<br>57±6.39 | 28.98±2.23<br>29.1±2.42<br>29.75±2.12 | 2 weeks | WOMAC, VAS |
| Santana ·2022 | Austria | RCT | Exercise<br>Placebo | 13<br>13 | 0/13<br>0/13 | 60±10.8<br>61±6.6 | 35.3±6.6<br>33.1±7.3 | 6 weeks | WOMAC |
| Sattari 2011 | Iran | RCT | Lateral edge insole<br>Knee brace<br>Placebo | 20<br>20<br>20 | –<br>–<br>– | 35-65<br>35-65<br>35-65 | –<br>–<br>– | 36 weeks | VAS |
| Sawitzke 2022 | US | RCT | Ultrasound<br>Placebo | 67<br>65 | 6/61<br>7/58 | 62.9±10.5<br>64.4±10.9 | 31.8±5.4<br>31.6±5.5 | 48 weeks | WOMAC, VAS |
| Schwartz 2023 | Israel | RCT | Lateral edge insole<br>Placebo | 26<br>12 | 9/17<br>8/4 | 67.5±8.8<br>64.6±8.0 | –<br>– | 12 weeks | WOMAC, VAS |
| Sedaghatnezhad 2019 | Iran | RCT | Exercise<br>Placebo | 15<br>15 | 1/14<br>4/11 | 53.8±7.43<br>59.6±7.43 | 26.55±2.08<br>27.7±1.93 | 20 days | VAS |
| Shah 2022 | Saudi Arabia | RCT | KT<br>Placebo | 20<br>20 | 13/7<br>14/6 | 55.55±3.80<br>55.3±3.88 | –<br>– | 4 weeks | WOMAC, VAS |
| Shen 2019 | China | RCT | Exercise<br>Placebo | 14<br>13 | 4/10<br>4/9 | 65.3±4.6<br>66.6±7.0 | 27.3±2.7<br>26.5±3.4 | 6 weeks | VAS |
| Silva 2007 | Brazil | RCT | Hydrotherapy<br>Exercise | 32<br>32 | 2/30<br>3/29 | 59±7.60<br>59±6.08 | –<br>– | 18 weeks | WOMAC, VAS |
| Siriratna 2022 | Thailand | RCT | HILT<br>Placebo | 21<br>21 | 3/18<br>5/16 | 66.1±9.4<br>65.0±8.5 | 28.1±5.2<br>27.4±5.8 | 12 weeks | WOMAC, VAS |
| Stausholm 2022 | Norway | RCT | LLLT<br>Placebo | 26<br>24 | 8/18<br>5/19 | 64.04±8.52<br>61.92±6.39 | 28.11±4.31<br>27.66±3.58 | 48 weeks | WOMAC, VAS |
| Tascioglu 2004 | Turkey | RCT | LLLT<br>Placebo | 20<br>20 | 5/15<br>7/13 | 59.92±7.59<br>64.27±10.55 | 28.63±6.48<br>29.56±9.54 | 24 weeks | WOMAC, VAS |
| Thoumie 2018 | France | RCT | knee brace<br>Placebo | 32<br>35 | 8/24<br>15/20 | 64.8±11.7<br>66.6±7.2 | 29.2±4.4<br>28.1±5.1 | 6 weeks | VAS |
| Uematsu 2021 | Japan | RCT | Ultrasound<br>Placebo | 37<br>33 | –<br>– | 67-78<br>68-78 | (23.88-28.44)<br>(21.92-26.94) | 12 weeks | VAS |
| Uysal 2020 | Turkey | RCT | ESWT<br>Placebo | 52<br>52 | 10/42<br>9/43 | 60.2±6.3<br>61.8±6.0 | 30.6±4.3<br>30.8±4.6 | 12 weeks | WOMAC, VAS |
| Vader 2020 | Canada | RCT | HILT<br>Placebo | 10<br>10 | 3/7<br>4/6 | 60.60±10.35<br>67.30±7.01 | –<br>– | 4 weeks | WOMAC, VAS |
| Vance 2012 | US | RCT | TENS<br>Placebo | 25<br>25 | 9/16<br>9/16 | 55±14.4<br>57±10.9 | 36.2±6.0<br>39.2±7.0 | – | VAS |
| Van 2010 | Netherlands | RCT | Lateral wedged insole<br>Knee brace | 45<br>46 | 20/25<br>11/35 | 54.4±6.5<br>54.9±7.4 | 29.4±4.9<br>29.0±4.2 | 24 weeks | WOMAC, VAS |
| Vassao 2019 | Brazil | RCT | LLLT<br>Exercise | 17<br>15 | 0/17<br>0/15 | 61.65±4.28<br>65.37±4.19 | 30.00±3.43<br>27.52±3.31 | 16 weeks | VAS |
| Vassao 2020 | Brazil | RCT | LLLT<br>Placebo | 17<br>16 | 0/17<br>0/16 | 61.65±4.42<br>61.19±4.45 | 30.00±3.53<br>30.44±4.76 | 8 weeks | WOMAC, VAS |

*(Continued)*

**Table 1.** (Continued)

| – | Country | Study design | Intervention | N | Sex (M/F) | Age (years) | BMI | Follow-up | Outcome |
|---|---------|--------------|--------------|---|-----------|-------------|-----|-----------|---------|
| Vassao 2021 | Brazil | RCT | LLLT<br>Exercise<br>Placebo | 13<br>13<br>10 | 0/13<br>0/13<br>0/10 | 62.29±4.39<br>61.57±4.42<br>66.5±4.06 | 30.11±3.64<br>30.49±4.35<br>27.24±2.99 | 8 weeks | WOMAC, VAS |
| Vincent 2020 | US | RCT | Exercise<br>Placebo | 19<br>17 | 7/12<br>5/12 | 66.8±5.4<br>68.6±7.1 | 28.7±6.6<br>32.8±18.2 | 16 weeks | VAS |
| Wageck 2016 | Australia | RCT | KT<br>Placebo | 38<br>38 | 3/35<br>7/31 | 69.6±6.9<br>68.66±6.3 | 30.0±4.9<br>31.3±4.1 | 19 days | WOMAC |
| Ye 2020 | China | RCT | Exercise<br>Placebo | 28<br>28 | 11/17<br>8/20 | 65.11±6.57<br>63.61±2.63 | 24.19±2.37<br>24.63±2.27 | 12 weeks | WOMAC |
| Yu 2016 | Australia | RCT | Knee brace<br>Placebo | 50<br>68 | 20/30<br>22/46 | 67.2±9.6<br>67.0±10.6 | 29.6±5.8<br>33.2±7.4 | 48 weeks | VAS |
| Yurtkuran 2007 | Turkey | RCT | LLLT<br>Placebo | 27<br>26 | 1/26<br>1/25 | 51.83±6.83<br>53.478±7.13 | 31.76±8.81<br>32.72±3.71 | 10 weeks | WOMAC, VAS |
| Zhang 2021 | China | RCT | ESWT<br>Placebo | 19<br>14 | 8/11<br>6/8 | 60.84±8.36<br>61.50±5.43 | 24.83±1.73<br>24.98±1.32 | 4 weeks | WOMAC, VAS |

Note: RCT = randomized controlled trial; LLLT = low level laser therapy; HILT = high intensity laser therapy; TENS = transcutaneous electrical nerve stimulation; IFC = interferential current; KT = kinesio taping; ESWT = extracorporeal shock wave therapy.

### 3.4 Inconsistency test

Fig 3 displays an inconsistency plot designed to assess heterogeneity among studies within the closed loops of the network meta-analysis. For WOMAC pain score, there were 23 closed loops, with IF ranging from 0.02 to 3.49. The majority of these closed loops had 95%CIs that contained 0, and only one closed loops of LLLT~HILT~exercise had 95%CIs approaching 0. Similarly, regarding WOMAC function score, there were 22 closed loops, with IF ranging from 1.17 to 15.98, and the majority of these closed loops had 95%CIs that contained 0. Only three closed loops had 95%CIs approaching 0, including loops of TENS~ultrasound~placebo, LLLT~HILT~exercise, and IFC~short wave diathermy~placebo. In terms of the WOMAC stiffness score, there were 15 closed loops, with IF ranging from 0.01 to 3.21, all of which were close to 0. Likewise, for the total WOMAC score, there were 10 closed loops, with IF ranging from 0.04 to 18.88, all of which were close to 0. The 95%CIs for these IF values contained 0, indicating no statistically significant differences. For VAS-rest, there were 31 closed loops, with IF ranging from 0.02 to 3.74, and the majority of these closed loops had 95%CIs that contained 0. Only two closed loops had 95%CIs approaching 0, including loops of lateral wedged insole~knee brace~placebo, and ultrasound~exercise~ESWT. And in terms of VAS-activity, there were only 3 closed loops, with IF ranging from 1.08 to 4.19. The majority of these closed loops had 95%CIs that contained 0, and only one closed loops of ultrasound~exercise~placebo had 95%CIs approaching 0. Overall, these results suggest that the data exhibited consistency.

### 3.5 Results of Network meta-analysis

#### 3.5.1 WOMAC pain score at last follow-up.
The results of the network meta-analysis revealed the following findings regarding WOMAC pain score at last follow-up: knee brace demonstrated a lower WOMAC pain score compared to lateral wedged insole. Exercise exhibited a lower WOMAC pain score than ultrasound and placebo. Other comparisons did not yield statistically significant differences (Fig 4 and Table 3).

A ranking graph depicting the distribution of probabilities for WOMAC pain score at last follow-up is presented in Fig 5. Based on the Surface Under the Cumulative Ranking Curve (SUCRA), knee brace obtained the lowest SUCRA rank, indicating the highest probability of relieving knee pain. Conversely, ultrasound had the lowest probability of relieving

**Table 2. Risk of bias of the included randomized controlled trials.**

| study | Sequence_generation | Allocation_concealment | Blinding | | | Selective_reporting-bias | Attrition_bias |
|---|---|---|---|---|---|---|---|
| | | | partici-pant | thera-pist | assessor | | |
| Adıgüzel 2021 | computer-generated random number list | unclear | no | no | yes | low risk | low risk |
| Ahmad 2023 | computer-generated randomization table | sealed opaque envelope | yes | no | yes | low risk | low risk |
| Ahmad 2023 | computer-generated randomization table | sealed opaque envelope | no | no | yes | low risk | low risk |
| Akaltun 2021 | envelope pulling procedure | unclear | yes | no | yes | low risk | low risk |
| Akyol 2010 | unclear | unclear | no | no | no | low risk | low risk |
| Alfredo 2020 | permuted block randomization method | unclear | yes | no | yes | low risk | low risk |
| Alfredo 2022 | computer-generated random number table | sealed opaque envelope | yes | yes | yes | low risk | low risk |
| Alghadir 2013 | unclear | sealed opaque envelope | no | no | yes | low risk | low risk |
| Ali 2014 | unclear | unclear | no | no | no | low risk | low risk |
| Alireza 2023 | unclear | sealed opaque envelope | no | no | yes | low risk | low risk |
| Alkhawajah 2019 | randomization procedure. | sealed opaque envelope | yes | no | yes | low risk | low risk |
| Allen 2021 | unclear | unclear | no | no | no | low risk | low risk |
| Alqualo-Costa 2021 | unclear | sealed opaque envelope | yes | no | yes | low risk | low risk |
| Rashoud 2014 | computer-generated randomization table | unclear | yes | no | yes | low risk | low risk |
| Amornthep 2023 | unclear | unclear | yes | no | yes | low risk | low risk |
| Anandkumar 2014 | unclear | sealed opaque envelope | yes | no | yes | low risk | low risk |
| Anwer 2014 | unclear | unclear | no | no | no | low risk | low risk |
| Artuç 2023 | computer-generated randomization table | unclear | yes | no | yes | low risk | low risk |
| Assar 2020 | unclear | unclear | no | no | yes | low risk | low risk |
| Atamaz 2012 | unclear | unclear | yes | yes | yes | low risk | low risk |
| Aydogdu 2017 | unclear | sealed opaque envelope | no | no | yes | low risk | low risk |
| Baykal 2023 | card drawing method | unclear | yes | no | yes | low risk | low risk |
| Bennell 2015 | computer-generated random number table | sealed opaque envelope | yes | no | yes | low risk | low risk |
| Bennell 2011 | random number table | unclear | yes | no | yes | low risk | low risk |
| Bruce-Brand 2012 | unclear | unclear | no | no | yes | low risk | low risk |
| Cantista 2020 | computer-generated randomization table | unclear | no | no | yes | low risk | low risk |
| Carpenedo 2021 | unclear | unclear | yes | no | yes | low risk | low risk |
| Chen 2014 | unclear | sealed opaque envelope | yes | yes | yes | low risk | low risk |
| Chen 2023 | computer-generated random number table | sealed opaque envelope | yes | yes | yes | low risk | low risk |
| Chen HX 2021 | computer-generated randomization table | unclear | no | no | yes | low risk | low risk |
| Chen PY 2021 | unclear | unclear | no | no | yes | low risk | low risk |
| Cherian 2016 | unclear | unclear | yes | no | yes | low risk | low risk |
| Cho 2015 | unclear | sealed opaque envelope | yes | yes | yes | low risk | low risk |
| Choi 2023 | unclear | unclear | no | no | yes | low risk | low risk |
| Danazumi 2021 | block randomization | unclear | yes | no | yes | low risk | low risk |
| Dantas 2023 | unclear | sealed opaque envelope | yes | no | yes | low risk | low risk |
| Dias 2016 | computer randomization | unclear | yes | no | yes | low risk | low risk |
| Dogan 2022 | unclear | sealed opaque envelope | yes | yes | yes | low risk | low risk |
| Donec 2020 | a computer-generated list. | sealed opaque envelope | yes | no | yes | low risk | low risk |

*(Continued)*

**Table 2.** (Continued)

| study | Sequence_generation | Allocation_concealment | Blinding | | | Selective_reporting-bias | Attrition_bias |
|---|---|---|---|---|---|---|---|
| | | | partici-pant | thera-pist | assessor | | |
| Ekici 2022 | computer-generated random number table | sealed opaque envelope | yes | no | yes | low risk | low risk |
| Elboim-Gabyzon 2023 | computer-generated random allocation | unclear | no | no | yes | low risk | low risk |
| Fazli 2023 | permuted block randomization | sealed opaque envelope | yes | no | yes | low risk | low risk |
| Fokmare 2023 | unclear | sealed opaque envelope | yes | no | yes | low risk | low risk |
| Foley 2014 | computer generated randomisation list | sealed opaque envelope | yes | no | yes | low risk | low risk |
| Foroughi 2011 | computerized randomization program | unclear | no | no | yes | low risk | low risk |
| Fukuda 2011 | unclear | sealed opaque envelope | yes | no | yes | low risk | low risk |
| Fukuda 2011 | unclear | sealed opaque envelope | yes | no | yes | low risk | low risk |
| Fung 2021 | unclear | unclear | yes | no | yes | low risk | low risk |
| Gao 2023 | computer-generated. | sealed opaque envelope | no | no | yes | low risk | low risk |
| Gholami 2023 | unclear | unclear | no | no | yes | low risk | low risk |
| Gomes 2020 | unclear | sealed opaque envelope | yes | no | yes | low risk | low risk |
| Günaydin 2022 | unclear | unclear | no | no | no | low risk | low risk |
| Guo 2021 | computer-generated. | unclear | yes | no | yes | low risk | low risk |
| Gur 2003 | unclear | unclear | yes | no | yes | low risk | low risk |
| Hammam 2020 | unclear | sealed opaque envelope | yes | no | yes | low risk | low risk |
| Han 2021 | computer-generated. | sealed opaque envelope | yes | yes | yes | low risk | low risk |
| Hinman 2003 | unclear | unclear | yes | no | yes | low risk | low risk |
| Ho 2022 | computer-generated random number table | unclear | yes | no | yes | low risk | low risk |
| Hu 2019 | computer-generated. | unclear | no | no | yes | low risk | low risk |
| Iijima 2020 | unclear | unclear | no | no | yes | low risk | low risk |
| Imamura 2016 | computer-generated. | sealed opaque envelope | yes | no | yes | low risk | low risk |
| Itoh 2008 | block randomised procedure | unclear | no | no | yes | low risk | low risk |
| Jang 2023 | block randomization technique | sealed opaque envelope | yes | no | yes | low risk | low risk |
| Jia 2022 | unclear | sealed opaque envelope | yes | no | yes | low risk | low risk |
| Jones 2013 | unclear | unclear | no | no | no | low risk | low risk |
| Jorge 2023 | computer-generated. | sealed opaque envelope | yes | no | yes | low risk | low risk |
| Karakas 2020 | block randomization method | sealed opaque envelope | yes | no | yes | low risk | low risk |
| Karimi 2021 | unclear | unclear | no | no | no | low risk | low risk |
| Kayamutlu 2016 | computer-generated. | unclear | yes | no | yes | low risk | low risk |
| Kayamutlu 2018 | computer-generated. | sealed opaque envelope | no | no | yes | low risk | low risk |
| Kheshie 2014 | specific identification number | unclear | yes | no | yes | low risk | low risk |
| Khosravi 2021 | unclear | unclear | no | no | yes | low risk | low risk |
| Khruakhorn 2021 | computer-generated. | sealed opaque envelope | yes | no | yes | low risk | low risk |
| Kilic 2020 | unclear | unclear | no | no | yes | low risk | low risk |
| Kitano 2023 | generated on a computer | sealed opaque envelope | yes | no | yes | low risk | low risk |
| Kocyigit 2015 | the numbered envelopes method | unclear | yes | no | yes | low risk | low risk |
| Laufer 2005 | unclear | sealed opaque envelope | yes | no | yes | low risk | low risk |
| Lee 2023 | random number table | unclear | yes | no | yes | low risk | low risk |
| Leon 2017 | unclear | sealed opaque envelope | yes | no | yes | low risk | low risk |

*(Continued)*

**Table 2.** (Continued)

| study | Sequence_generation | Allocation_concealment | Blinding | | | Selective_reporting-bias | Attrition_bias |
|---|---|---|---|---|---|---|---|
| | | | partici-pant | thera-pist | assessor | | |
| Lewinson 2016 | computer-generated. | sealed opaque envelope | yes | no | yes | low risk | low risk |
| Liao 2020 | random number table | sealed opaque envelope | yes | no | yes | low risk | low risk |
| Lin 2022 | unclear | unclear | no | no | yes | low risk | low risk |
| Maheu 2022 | computer-generated. | sealed opaque envelope | yes | no | yes | low risk | low risk |
| Mahler 2018 | stratified block randomisation | sealed opaque envelope | yes | no | yes | low risk | low risk |
| Marconcin 2021 | unclear | sealed opaque envelope | no | no | yes | low risk | low risk |
| Mascarin 2012 | unclear | unclear | yes | no | yes | low risk | low risk |
| McManus 2021 | unclear | unclear | no | no | no | low risk | low risk |
| Messier 2022 | random permuted-block randomization | unclear | no | no | yes | low risk | low risk |
| Messier 2022 | random permuted-block randomization | unclear | no | no | yes | low risk | low risk |
| Mete 2022 | computer-generated | unclear | no | no | yes | low risk | low risk |
| Mobina 2019 | unclear | unclear | yes | no | yes | low risk | low risk |
| Mohamed 2022 | unclear | unclear | yes | no | yes | low risk | low risk |
| MohammedSadiq 2021 | computer-generated | unclear | yes | no | yes | low risk | low risk |
| Mostafa 2021 | unclear | sealed opaque envelope | yes | no | yes | low risk | low risk |
| Müller-Rath 2011 | unclear | unclear | no | no | yes | low risk | low risk |
| Nambi 2016 | unclear | sealed opaque envelope | yes | no | yes | low risk | low risk |
| Nazari 2018 | computer-generated | unclear | yes | yes | yes | low risk | low risk |
| Oğuz 2021 | unclear | sealed opaque envelope | no | no | yes | low risk | low risk |
| Palmer 2014 | unclear | sealed opaque envelope | yes | no | yes | low risk | low risk |
| Park 2021 | random number tables | unclear | no | no | yes | low risk | low risk |
| Pierosimone 2020 | unclear | sealed opaque envelope | yes | no | yes | low risk | low risk |
| Pinto 2020 | computer-generated | unclear | yes | no | yes | low risk | low risk |
| Pozsgai 2022 | unclear | sealed opaque envelope | yes | no | yes | low risk | low risk |
| Qiestad 2023 | Computer-generated randomization lists | unclear | yes | no | yes | low risk | low risk |
| Rabiei 2023 | computer-generated | sealed opaque envelope | no | no | yes | low risk | low risk |
| Rafiq 2021 | computer-generated | unclear | no | no | yes | low risk | low risk |
| Rahlf 2017 | unclear | sealed opaque envelope | yes | no | yes | low risk | low risk |
| Rego 2023 | unclear | sealed opaque envelope | no | no | yes | low risk | low risk |
| Reichenbach 2021 | computer-generated | unclear | no | no | yes | low risk | low risk |
| Rewald 2019 | computer-generated | unclear | no | no | yes | low risk | low risk |
| Ridvan 2020 | unclear | unclear | yes | no | yes | low risk | low risk |
| Robbins 2021 | computer-generated | unclear | yes | no | yes | low risk | low risk |
| Samaan 2022 | computer-generated randomized table | unclear | no | no | yes | low risk | low risk |
| Santana 2022 | unclear | unclear | no | no | yes | low risk | low risk |
| Sattari 2011 | computer-generated procedure | unclear | no | no | yes | low risk | low risk |
| Sawitzke 2022 | random block sizes | unclear | yes | no | yes | low risk | low risk |
| Schwartz 2023 | unclear | unclear | yes | no | yes | low risk | low risk |
| Sedaghatnezhad 2019 | flipping a coin | unclear | yes | no | yes | low risk | low risk |
| Shah 2022 | computer-generated | sealed opaque envelope | yes | no | yes | low risk | low risk |
| Shen 2019 | computer-generated | sealed opaque envelope | no | no | yes | low risk | low risk |
| Silva 2007 | drawing lots | unclear | no | no | yes | low risk | low risk |

*(Continued)*

**Table 2.** (Continued)

| study | Sequence_generation | Allocation_concealment | Blinding | | | Selective_ reporting-bias | Attrition_ bias |
|---|---|---|---|---|---|---|---|
| | | | partici-pant | thera-pist | assessor | | |
| Siriratna 2022 | unclear | sealed opaque envelope | no | no | yes | low risk | low risk |
| Stausholm 2022 | unclear | sealed opaque envelope | yes | no | yes | low risk | low risk |
| Tascioglu 2004 | numbered envelopes | unclear | yes | no | yes | low risk | low risk |
| Thoumie 2018 | unclear | unclear | no | no | yes | low risk | low risk |
| Uematsu 2021 | unclear | sealed opaque envelope | yes | yes | yes | low risk | low risk |
| Uysal 2020 | unclear | unclear | no | no | yes | low risk | low risk |
| Vader 2020 | computer-generated | sealed opaque envelope | yes | no | yes | low risk | low risk |
| Vance 2012 | unclear | sealed opaque envelope | yes | no | yes | low risk | low risk |
| Van 2010 | Computer-generated procedure | unclear | no | no | yes | low risk | low risk |
| Vassao 2019 | a random table of numbers | unclear | yes | no | yes | low risk | low risk |
| Vassao 2020 | computer-generated | sealed opaque envelope | yes | no | yes | low risk | low risk |
| Vassao 2021 | random table of numbers | unclear | yes | no | yes | low risk | low risk |
| Vincent 2020 | computer-generated | sealed opaque envelope | no | no | yes | low risk | low risk |
| Wageck 2016 | unclear | sealed opaque envelope | no | no | yes | low risk | low risk |
| Ye 2020 | computer-generated | sealed opaque envelope | no | no | yes | low risk | low risk |
| Yu 2016 | unclear | unclear | no | no | yes | low risk | low risk |
| Yurtkuran 2007 | computer-generated | unclear | yes | no | yes | low risk | low risk |
| Zhang 2021 | computer-generated | sealed opaque envelope | yes | no | yes | low risk | low risk |

knee pain. The SUCRA rankings for WOMAC pain score at last follow-up were as follows: knee brace (18.7%) <exercise (22.8%) <HILT (25.3%) <ESWT (31.0%) <hydrotherapy (31.8%) <LLLT (49.4%) <KT (53.3%) <TENS (56.9%) <short wave diathermy (63.7%) <IFC (63.9%) <lateral wedged insole (71.7%) <placebo (78.8%) <ultrasound (82.6%).

**3.5.2 WOMAC function score at last follow-up.** The results of the network meta-analysis indicated the following findings concerning WOMAC function score at last follow-up: knee brace demonstrated a lower WOMAC function score compared to LLLT, HILT, TENS, IFC, short wave diathermy, ultrasound, lateral wedged insole, exercise, hydrotherapy, KT, ESWT and placebo. Hydrotherapy exhibited lower WOMAC function score compared to LLLT, TENS and ultrasound. Exercise showed lower WOMAC function score compared to ultrasound. Other comparisons did not yield statistically significant differences (**Fig 4** and **Table 3**).

A ranking graph depicting the distribution of probabilities for WOMAC function score at last follow-up is presented in **Fig 5**. Based on the SUCRA, knee brace obtained the lowest SUCRA rank, indicating the highest probability of recovering knee function. Conversely, ultrasound had the lowest probability of recovering knee function. The SUCRA rankings for WOMAC function score at last follow-up were as follows: knee brace (0.1%) <hydrotherapy (12%) <ESWT (29.5%) <exercise (32.6%) <HILT (42.8%) <IFC (51%) <short wave diathermy (51.5%) <TENS (57.7%) <LLLT (58.7%) <KT (72.3%) <lateral wedged insole (72.4%) <placebo (78.7%) <ultrasound (90.7%).

**3.5.3 WOMAC stiffness score at last follow-up.** The results of the network meta-analysis indicated the following findings concerning WOMAC stiffness score at last follow-up: knee brace demonstrated a lower WOMAC stiffness score compared to LLLT, HILT, TENS, IFC, short wave diathermy, ultrasound, lateral wedged insole, exercise, hydrotherapy, KT, ESWT and placebo. Exercise showed lower WOMAC stiffness score compared to placebo. Other comparisons did not yield statistically significant differences (**Fig 4** and **Table 3**).

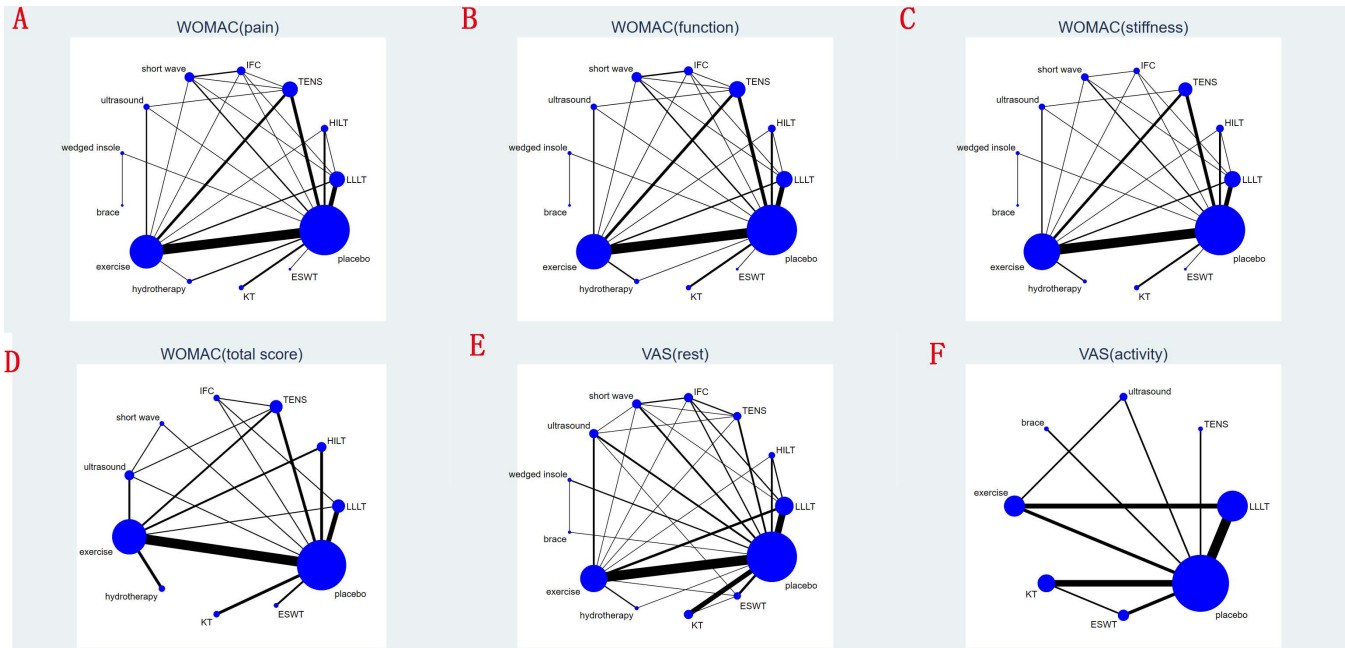

**Fig 2. Network analysis of eligible comparison for (A) WOMAC pain score, (B) WOMAC function score, (C) WOMAC stiffness score, (D) total WOMAC score, (E) VAS-rest and (F) VAS-activity at last follow-up.** The size of each node represents the number of participants, while the thickness of the line represents the number of studies directly comparing the two interventions.

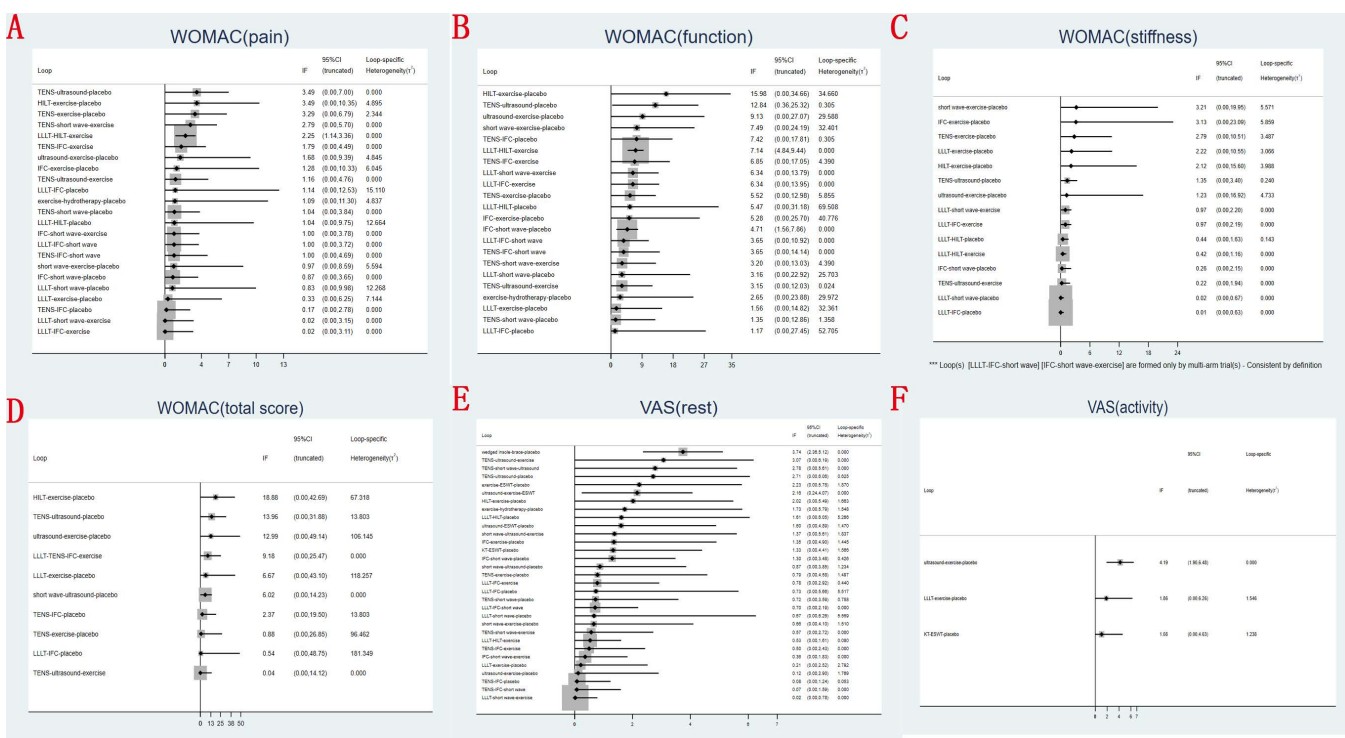

**Fig 3. Inconsistency plot of eligible comparison for (A) WOMAC pain score, (B) WOMAC function score, (C) WOMAC stiffness score, (D) total WOMAC score, (E) VAS-rest and (F) VAS-activity at last follow-up.**

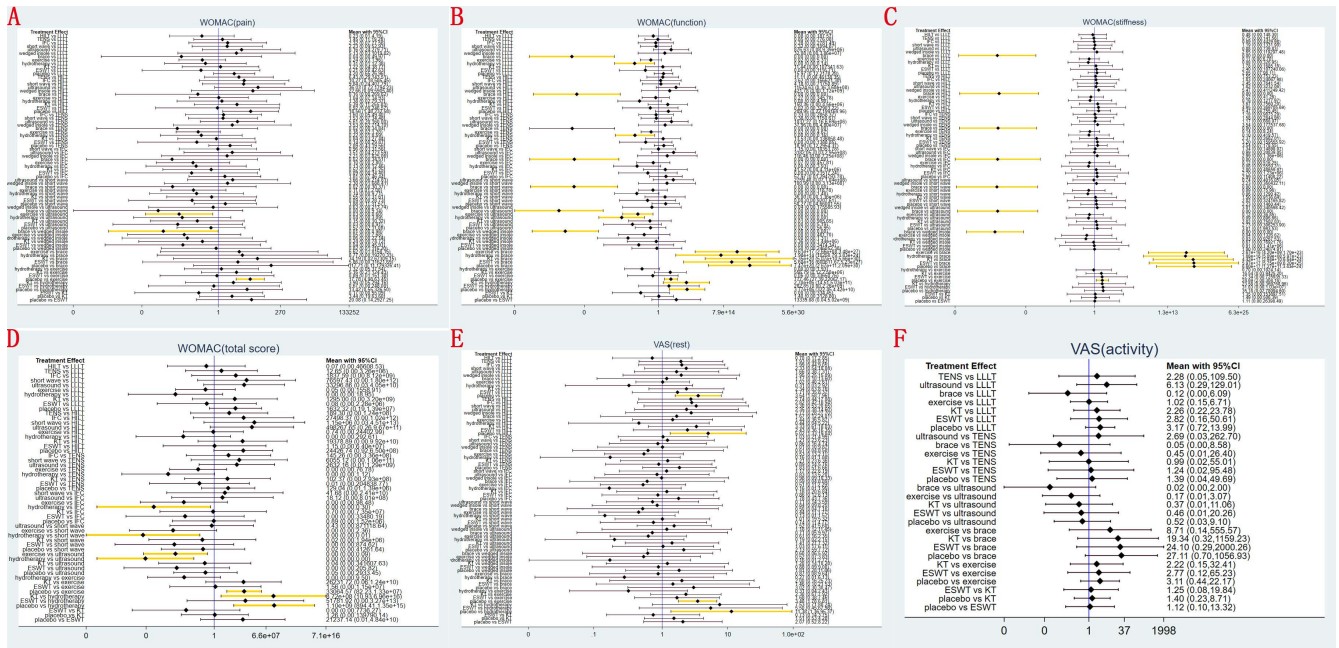

**Fig 4. Forest plots for (A) WOMAC pain score, (B) WOMAC function score, (C) WOMAC stiffness score, (D) total WOMAC score, (E) VAS-rest and (F) VAS-activity at last follow-up.**

A ranking graph depicting the distribution of probabilities for WOMAC stiffness score at last follow-up is presented in **Fig 5**. Based on the SUCRA, knee brace obtained the lowest SUCRA rank, indicating the highest probability of relieving knee stiffness. Conversely, placebo had the lowest probability of relieving knee stiffness. The SUCRA rankings for WOMAC stiffness score at last follow-up were as follows: knee brace (0%) <exercise (30.6%) <hydrotherapy (36%) <HILT (48.9%) <TENS (53.7%) <IFC (55.4%) <ultrasound (55.4%) <LLLT (57%) <short wave diathermy (57.9%) <ESWT (61.3%) <KT (61.5%) <lateral wedged insole (62.4%) <placebo (70.1%).

**3.5.4 Total WOMAC score at last follow-up.** The results of the network meta-analysis indicated the following findings concerning total WOMAC score at last follow-up: hydrotherapy demonstrated a lower total WOMAC score compared to IFC, short wave diathermy, ultrasound, KT and placebo. Exercise showed lower total WOMAC score compared to ultrasound and placebo. Other comparisons did not yield statistically significant differences (**Fig 4** and **Table 3**).

A ranking graph depicting the distribution of probabilities for total WOMAC score at last follow-up is presented in **Fig 5**. Based on the SUCRA, hydrotherapy obtained the lowest SUCRA rank, indicating the highest probability. Conversely, short wave diathermy had the lowest probability. The SUCRA rankings for total WOMAC score at last follow-up were as follows: hydrotherapy (4.1%) <exercise (25.1%) <HILT (28.5%) <ESWT (31.4%) <LLLT (38.3%) <TENS (49.7%) <KT (68.7%) <IFC (69.2%) <placebo (72.6%) <ultrasound (80.9%) <short wave diathermy (91.4%).

**3.5.5 VAS-rest at last follow-up.** The results of the network meta-analysis indicated the following findings concerning VAS-rest at last follow-up: Placebo demonstrated a higher VAS-rest compared to hydrotherapy, HILT, LLLT, and exercise. Other comparisons did not yield statistically significant differences (**Fig 4** and **Table 3**).

A ranking graph depicting the distribution of probabilities for VAS-rest at last follow-up is presented in **Fig 5**. Based on the SUCRA, hydrotherapy obtained the lowest SUCRA rank, indicating the highest probability. Conversely, placebo had the lowest probability. The SUCRA rankings for VAS-rest at last follow-up were as follows: hydrotherapy (10.6%) <HILT (23%) <LLLT (31.9%) <exercise (32.2%) <knee brace (43.1%) <ultrasound (53.7%) <ESWT (54.5%)

<lateral wedged insole (59.1%) <TENS (59.5%) <IFC (60.7%) <short wave diathermy (67.3%) <KT (67.8%) <placebo (86.6%).

### 3.5.6 VAS-activity at last follow-up.

The results of the network meta-analysis indicated that all comparisons did not yield statistically significant differences concerning VAS-activity at last follow-up (**Fig 4** and **Table 3**).

A ranking graph depicting the distribution of probabilities for VAS-activity at last follow-up is presented in **Fig 5**. Based on the SUCRA, knee brace obtained the lowest SUCRA rank, indicating the highest probability. Conversely, ultrasound had the lowest probability. The SUCRA rankings for VAS-activity at last follow-up were as follows: knee brace (9.4%) <LLLT (33.5%) <exercise (35.6%) <TENS (55.9%) <KT (56.8%) <ESWT (62.1%) <placebo (69.2%) <ultrasound (77.4%).

## 3.6 Publication bias

Based on the outcomes observed for WOMAC pain score, WOMAC function score, WOMAC stiffness score, total WOMAC score, VAS-rest and VAS-activity at last follow-up, network meta-analysis showed that the corrected funnel plots were generated to assess publication bias and potential small sample effects. The analysis revealed that most data points were well-distributed within the funnel plot, displaying relative symmetry on both sides. Additionally, the regression line closely paralleled the X-axis, indicating minimal likelihood of publication bias or small sample effects (**Fig 6**).

## 4 Discussion

Despite evidence from numerous clinical randomized controlled trials and meta-analyses that physical therapy has promising effects on knee osteoarthritis [7–17], a network meta-analysis comprehensively analyzing the clinical efficacy of 12 therapeutic options, including low level laser therapy, high intensity laser therapy, transcutaneous electrical nerve stimulation, interferential current, short wave diathermy, ultrasound, lateral wedged insole, knee brace, exercise, hydrotherapy, kinesio taping and extracorporeal shock wave therapy, is currently lacking. This implies that while we recognize the positive role of physical therapy in improving symptoms and functionality in knee osteoarthritis patients, these methods may differ in mechanisms of action, applicability, and efficacy variances. For instance, LLLT may function by promoting cellular metabolism and tissue repair, whereas HLLT may focus more on alleviating inflammation and pain in deep tissues. TENS works by modulating neural conduction to reduce pain perception. However, due to the absence of more in-depth and comprehensive network meta-analysis, it remains unclear which physical therapy method is most effective for different patient populations and how to optimize the combination of these interventions for the best therapeutic outcomes. This study aims to ascertain the comparative effects of various physical therapies for knee osteoarthritis, aiding clinicians in precisely selecting the most suitable physical therapy method based on individual patient conditions, enhancing treatment efficacy, reducing unnecessary medical resource wastage, and providing robust guidance and evidence for future research directions. Based on 139 included randomized controlled trials, we conducted a focused analysis and discussion of WOMAC and VAS scores. Overall results indicate that knee brace are the most effective, followed by hydrotherapy, exercise, HILT, ESWT, LLLT, TENS, IFC, KT, short wave diathermy, ultrasound, and lateral wedged insole. When considering pain relief in particular, the hierarchy is hydrotherapy, HILT, LLLT, exercise, knee brace, ultrasound, ESWT, lateral wedged insole, TENS, IFC, short wave diathermy, and KT.

On the whole, Knee orthoses provide the most effective treatment for knee osteoarthritis, with their primary mechanisms of action being as follows: (1) Improving joint biomechanics by adjusting the knee joint's force line to evenly distribute load and reduce excessive stress on cartilage and soft tissues. Knee orthoses can apply a valgus force through a brace or modify the ground reaction force to change the medial knee load [18]. They rapidly enhance the knee's walking patterns and biomechanical gait efficiency [19–21]. Cudejko [22] and colleagues propose that knee orthoses widen the medial joint space during walking, addressing a primary cause of pain symptoms. (2) Enhancing joint stability by limiting excessive knee movement in unstable conditions, thereby reducing injury risk and pain [23,24]. (3) Alleviating muscle fatigue by supporting surrounding muscles and reducing their workload to maintain joint function. (4) Adjusting

*(Continued)*

**Table 3. Comparative primary outcomes for WOMAC pain score, WOMAC function score, WOMAC stiffness score, total WOMAC score, VAS-rest and VAS-activity. Significant results are in bold text.**

| | LLLT | HILT | TENS | IFC | short wave diathermy | ultrasound |
|---|---|---|---|---|---|---|
| **WOMAC pain score** | LLLT | 0.23 (0.01,4.19) | 1.45 (0.11,19.28) | 2.32 (0.07,77.56) | 2.23 (0.09,52.93) | 8.16 (0.24,279.71) |
| **WOMAC function score** | 13.15 (0.01,32425.79) | HILT | 6.43 (0.29,143.51) | 10.27 (0.18,585.45) | 9.84 (0.24,407.79) | 36.03 (0.72,1794.23) |
| | 1.18 (0.00,1086.82) | 0.09 (0.00,373.76) | TENS | 1.60 (0.05,49.40) | 1.53 (0.07,34.48) | 5.61 (0.20,160.00) |
| | 3.63 (0.00,36657.10) | 0.28 (0.00,12702.52) | 3.07 (0.00,26899.59) | IFC | 0.96 (0.03,31.56) | 3.51 (0.04,273.59) |
| | 3.15 (0.00,10851.37) | 0.24 (0.00,4113.66) | 2.66 (0.00,8143.64) | 0.87 (0.00,7786.47) | short wave diathermy | 3.66 (0.06,214.81) |
| | 0.00 (0.00,12.22) | 0.00 (0.00,2.77) | 0.00 (0.00,6.22) | 0.00 (0.00,29.12) | 0.00 (0.00,13.62) | ultrasound |
| | 0.03 (0.00,37523.63) | 0.00 (0.00,6311.61) | 0.03 (0.00,33320.21) | 0.01 (0.00,53421.83) | 0.01 (0.00,30727.87) | 27.32 (0.00,1.31e+08) |
| | **4.68e+19 (9.66e+09,2.27e+29)** | **3.56e+18 (4.43e+08,2.86e+28)** | **3.96e+19 (7.91e+09,1.98e+29)** | **1.29e+19 (9.16e+08,1.81e+29)** | **1.49e+19 (1.67e+09,1.32e+29)** | **4.10e+22 (3.47e+12,4.85e+32)** |
| | 48.69 (0.19,12590.73) | 3.70 (0.00,4741.49) | 41.14 (0.21,7899.52) | 13.41 (0.00,82345.77) | 15.47 (0.01,28408.83) | 42676.09 (15.25,1.19e+08) |
| | **236290.22 (6.94,8.04e+09)** | 17971.55 (0.20,1.58e+09) | **199673.19 (6.41,6.22e+09)** | 65088.72 (0.25,1.66e+10) | 75055.97 (0.67,8.36e+09) | **2.07e+08 (1294.96,3.31e+13)** |
| | 0.09 (0.00,820.79) | 0.01 (0.00,201.68) | 0.07 (0.00,747.27) | 0.02 (0.00,2387.19) | 0.03 (0.00,1048.55) | 76.65 (0.00,5.32e+06) |
| | 835.40 (0.00,7.79e+08) | 63.54 (0.00,1.33e+08) | 705.94 (0.00,6.92e+08) | 230.12 (0.00,1.14e+09) | 265.36 (0.00,6.48e+08) | 732250.27 (0.19,2.78e+12) |
| | 0.06 (0.00,8.54) | 0.00 (0.00,4.52) | 0.05 (0.00,8.27) | 0.02 (0.00,87.72) | 0.02 (0.00,26.12) | 54.89 (0.02,171585.85) |
| | **LLLT** | **HILT** | **TENS** | **IFC** | **short wave diathermy** | **ultrasound** |
| **WOMAC stiffness score** | LLLT | 0.48 (0.00,148.30) | 0.75 (0.00,141.56) | 0.89 (0.00,4429.98) | 1.19 (0.00,1331.90) | 0.88 (0.00,739.83) |
| **Total WOMAC score** | 14.96 (0.00,1.04e+07) | HILT | 1.55 (0.00,753.23) | 1.83 (0.00,24067.98) | 2.45 (0.00,7641.54) | 1.82 (0.00,3232.06) |
| | 0.08 (0.00,20352.80) | 0.01 (0.00,3453.01) | TENS | 1.18 (0.00,9875.29) | 1.58 (0.00,2844.96) | 1.17 (0.00,668.97) |
| | 0.00 (0.00,2405.14) | 0.00 (0.00,1353.35) | 0.01 (0.00,15916.11) | IFC | 1.34 (0.00,14898.07) | 0.99 (0.00,20839.05) |
| | 0.00 (0.00,307.41) | 0.00 (0.00,34.35) | 0.00 (0.00,2892.01) | 0.02 (0.00,1.39e+07) | short wave diathermy | 0.74 (0.00,4016.05) |
| | 0.00 (0.00,36.51) | 0.00 (0.00,3.90) | 0.00 (0.00,186.89) | 0.06 (0.00,2.44e+06) | 2.30 (0.00,4.61e+06) | ultrasound |
| | | | | | | |
| | | | | | | |
| | 20.26 (0.00,639640.79) | 1.35 (0.00,44713.44) | 256.24 (0.01,5.04e+06) | 37222.39 (0.01,1.36e+11) | 1.55e+06 (0.42,5.67e+12) | **674466.06 (10.58,4.30e+10)** |
| | 673702.35 (0.05,8.60e+12) | 45020.33 (0.00,5.93e+11) | 8.52e+06 (0.89,8.15e+13) | **1.24e+09 (3.36,4.55e+17)** | **5.16e+10 (141.26,1.89e+19)** | **2.24e+10 (1125.11,4.47e+17)** |

| lateral wedged insole | knee brace | exercise | hydrotherapy | KT | ESWT | placebo |
|---|---|---|---|---|---|---|
| 5.13 (0.03,1019.82) | 0.04 (0.00,49.97) | 0.24 (0.03,1.96) | 0.31 (0.01,12.38) | 1.22 (0.04,38.77) | 0.21 (0.00,42.63) | 4.20 (0.65,26.96) |
| 22.66 (0.09,5905.40) | 0.16 (0.00,269.62) | 1.04 (0.07,14.93) | 1.38 (0.02,79.37) | 5.39 (0.11,255.83) | 0.92 (0.00,246.57) | 18.56 (1.48,232.58) |
| 3.53 (0.02,714.70) | 0.02 (0.00,34.84) | 0.16 (0.02,1.20) | 0.22 (0.01,8.59) | 0.84 (0.03,27.44) | 0.14 (0.00,29.87) | 2.89 (0.43,19.56) |
| 2.21 (0.01,825.06) | 0.02 (0.00,34.51) | 0.10 (0.00,2.80) | 0.13 (0.00,12.46) | 0.52 (0.01,41.25) | 0.09 (0.00,34.40) | 1.81 (0.07,46.44) |
| 2.30 (0.01,688.41) | 0.02 (0.00,30.37) | 0.11 (0.01,2.00) | 0.14 (0.00,9.71) | 0.55 (0.01,31.61) | 0.09 (0.00,28.73) | 1.88 (0.11,31.57) |
| 0.63 (0.00,213.74) | 0.00 (0.00,9.14) | **0.03 (0.00,0.60)** | 0.04 (0.00,3.09) | 0.15 (0.00,10.32) | 0.03 (0.00,8.91) | 0.52 (0.02,11.08) |
| lateral wedged insole | **0.01 (0.00,0.98)** | 0.05 (0.00,7.68) | 0.06 (0.00,22.14) | 0.24 (0.00,74.72) | 0.04 (0.00,45.91) | 0.82 (0.01,116.20) |
| **1.50e+21 (4.33e+13,5.20e+28)** | knee brace | 6.62 (0.01,8158.15) | 8.77 (0.00,19270.33) | 34.19 (0.02,67309.15) | 5.86 (0.00,31621.69) | 117.71 (0.11,129326.41) |
| 1561.88 (0.00,1.18e+09) | **9.61e+17 (2.66e+08,3.48e+27)** | exercise | 1.32 (0.05,37.54) | 5.16 (0.21,124.43) | 0.89 (0.01,151.38) | **17.78 (4.98,63.45)** |
| 7.58e+06 (0.76,7.55e+13) | **1.98e+14 (10250.79,3.83e+24)** | 4853.19 (0.52,4.55e+07) | hydrotherapy | 3.90 (0.05,294.16) | 0.67 (0.00,248.00) | 13.42 (0.55,326.50) |
| 2.81 (0.00,1.13e+07) | **5.35e+20 (5.07e+10,5.66e+30)** | 0.00 (0.00,8.06) | 0.00 (0.00,0.07) | KT | 0.17 (0.00,55.09) | 3.44 (0.19,63.60) |
| 26799.27 (0.00,2.50e+12) | **5.60e+16 (600611.67,5.23e+27)** | 17.16 (0.00,9.91e+06) | 0.00 (0.00,28194.29) | 9553.50 (0.00,3.05e+10) | ESWT | 20.08 (0.14,2927.25) |
| 2.01 (0.00,993046.38) | **7.47e+20 (2.67e+11,2.09e+30)** | 0.00 (0.00,0.04) | 3.77e+06 (322.45,4.42e+10) | 0.72 (0.00,1604.72) | 0.00 (0.00,28.19) | placebo |
| lateral wedged insole | knee brace | exercise | hydrotherapy | KT | ESWT | placebo |
| 2.65 (0.00,119297.48) | **2.49e+17 (3.16e+10,1.96e+24)** | 0.11 (0.00,6.70) | 0.08 (0.00,326.95) | 1.78 (0.00,1685.16) | 2.40 (0.00,107240.06) | 2.65 (0.07,98.11) |
| 5.47 (0.00,411249.42) | **1.21e+17 (1.08e+10,1.35e+24)** | 0.22 (0.00,41.26) | 0.16 (0.00,1217.97) | 3.67 (0.00,7560.20) | 4.95 (0.00,369705.69) | 5.47 (0.04,755.45) |
| 3.54 (0.00,177537.68) | **1.86e+17 (2.20e+10,1.58e+24)** | 0.14 (0.00,8.24) | 0.10 (0.00,419.57) | 2.37 (0.00,2662.01) | 3.20 (0.00,159555.92) | 3.54 (0.07,178.55) |
| 2.99 (0.00,1.36e+06) | **2.21e+17 (5.41e+09,9.01e+24)** | 0.12 (0.00,516.25) | 0.08 (0.00,5550.31) | 2.00 (0.00,48659.97) | 2.70 (0.00,1.23e+06) | 2.99 (0.00,11408.22) |
| 2.23 (0.00,360432.11) | **2.95e+17 (1.54e+10,5.65e+24)** | 0.09 (0.00,75.96) | 0.06 (0.00,1290.42) | 1.50 (0.00,9136.64) | 2.02 (0.00,324165.82) | 2.23 (0.00,1460.44) |
| 3.01 (0.00,340598.42) | **2.19e+17 (1.47e+10,3.26e+24)** | 0.12 (0.00,36.85) | 0.09 (0.00,896.66) | 2.02 (0.00,7502.03) | 2.73 (0.00,306253.90) | 3.01 (0.01,993.53) |
| lateral wedged insole | **6.59e+17 (5.36e+12,8.11e+22)** | 0.04 (0.00,1326.62) | 0.03 (0.00,9267.53) | 0.67 (0.00,76921.76) | 0.91 (0.00,1.41e+06) | 1.00 (0.00,24074.91) |
| | knee brace | **2.67e+16 (4.20e+09,1.70e+23)** | **1.88e+16 (5.89e+08,5.97e+23)** | **4.42e+17 (2.94e+10,6.64e+24)** | **5.97e+17 (5.75e+09,6.20e+25)** | **6.60e+17 (1.27e+11,3.43e+24)** |
| | | exercise | 0.70 (0.00,1024.14) | 16.54 (0.03,9425.20) | 22.34 (0.00,725935.33) | **24.69 (2.00,304.16)** |
| | | 33259.16 (0.11,1.05e+10) | hydrotherapy | 23.58 (0.00,369768.96) | 31.83 (0.00,1.03e+07) | 35.18 (0.02,78094.60) |

*(Continued)*

**Table 3.** (Continued)

| | LLLT | HILT | TENS | IFC | short wave diathermy | ultrasound |
|---|---|---|---|---|---|---|
| | 0.00 (0.00,1908.50) | 0.00 (0.00,264.20) | 0.01 (0.00,27972.20) | 1.42 (0.00,1.48e+08) | 59.15 (0.00,6.77e+09) | 25.71 (0.00,2.26e+08) |
| | 13.01 (0.00,3.87e+08) | 0.87 (0.00,4.84e+07) | 164.58 (0.00,5.55e+09) | 23907.68 (0.00,1.91e+13) | 996561.45 (0.00,8.69e+14) | 433204.75 (0.00,3.86e+13) |
| | 0.00 (0.00,5.23) | 0.00 (0.00,1.09) | 0.01 (0.00,83.22) | 1.13 (0.00,1.93e+06) | 46.93 (0.00,9.09e+07) | 20.40 (0.00,1.22e+06) |
| | LLLT | HILT | TENS | IFC | short wave diathermy | ultrasound |
| VAS-rest | LLLT | 0.70 (0.17,2.85) | 1.93 (0.44,8.42) | 1.99 (0.44,9.05) | 2.33 (0.54,10.04) | 1.66 (0.38,7.31) |
| VAS-activity | | HILT | 2.74 (0.44,17.08) | 2.82 (0.42,18.94) | 3.30 (0.53,20.65) | 2.35 (0.38,14.60) |
| | 0.44 (0.01,21.04) | | TENS | 1.03 (0.21,4.99) | 1.21 (0.23,6.23) | 0.86 (0.16,4.74) |
| | | | | IFC | 1.17 (0.22,6.19) | 0.83 (0.13,5.20) |
| | | | | | short wave diathermy | 0.71 (0.14,3.59) |
| | 0.16 (0.01,3.44) | | 0.37 (0.00,36.41) | | | ultrasound |
| | | | | | | |
| | 8.54 (0.16,444.61) | | 19.49 (0.12,3258.42) | | | 52.35 (0.50,5483.52) |
| | 0.98 (0.15,6.46) | | 2.24 (0.04,132.28) | | | 6.01 (0.33,111.14) |
| | | | | | | |
| | 0.44 (0.04,4.64) | | 1.01 (0.02,55.86) | | | 2.71 (0.09,81.03) |
| | 0.35 (0.02,6.36) | | 0.81 (0.01,62.44) | | | 2.17 (0.05,95.62) |
| | 0.32 (0.07,1.39) | | 0.72 (0.02,25.68) | | | 1.93 (0.11,33.94) |

proprioception to improve patients' awareness of knee joint position and movement, enhancing joint control [25]. (5) Reducing the inflammatory response by limiting inflammation spread and easing related symptoms.

For the reduction of pain, this study demonstrates that aquatic therapy is particularly effective. This therapy, which can involve exercise in water or the use of water's properties for treatment, has been shown by evidence from past studies to significantly reduce pain and enhance physical function through heat stimulation and buoyancy. Water temperatures ranging from 33.5°C to 35.5°C are optimal for allowing prolonged immersion without thermal discomfort and for providing an adequate exercise duration for therapeutic benefits [26–28]. Increased water depth provides greater buoyancy, unloading the joints and consequently easing the pain associated with knee osteoarthritis [29]. Additionally, aquatic therapy addresses multiple psychosocial factors, including depression, self-efficacy, and exercise avoidance [30]. According to Lim's 2010 research, significant improvements were observed in the SF-36 health survey scores for individuals with knee osteoarthritis following aquatic therapy [31]. Aquatic therapy stands out in pain management, possibly due to the reduced impact and increased comfort of exercising in water compared to land [32].

Both high-energy and low-energy laser therapies exhibit pronounced therapeutic effects for the management of knee osteoarthritis (KOA). This study reveals that high-energy laser therapy, in terms of overall efficacy, is surpassed only by knee orthoses, hydrotherapy, and exercise therapy, with low-energy laser therapy following extracorporeal shockwave therapy. In terms of pain alleviation, the efficacy of both high-energy and low-energy laser therapies is secondary only to that of hydrotherapy. Owing to its non-invasive mechanical action in therapy, low-level laser therapy has emerged as a primary treatment option for a variety of musculoskeletal pain conditions. Clinically, it has been shown to mitigate pain and inflammation, facilitate healing and tissue repair, and enhance blood circulation [33,34]. High-intensity laser therapy,

| lateral wedged insole | knee brace | exercise | hydrotherapy | KT | ESWT | placebo |
|---|---|---|---|---|---|---|
| | | 0.00 (0.00,18.01) | 0.00 (0.00,0.09) | KT | 1.35 (0.00,153897.51) | 1.49 (0.00,506.39) |
| | | 0.64 (0.00,4.75e+06) | 0.00 (0.00,12192.51) | 16848.45 (0.00,2.20e+12) | ESWT | 1.11 (0.00,26398.49) |
| | | 0.00 (0.00,0.01) | 0.00 (0.00,0.00) | 0.79 (0.00,87984.00) | 0.00 (0.00,107.40) | placebo |
| lateral wedged insole | knee brace | exercise | hydrotherapy | KT | ESWT | placebo |
| 1.95 (0.25,15.09) | 1.17 (0.10,13.60) | 1.02 (0.40,2.61) | 0.31 (0.03,2.92) | 2.34 (0.63,8.76) | 1.71 (0.35,8.37) | **3.54 (1.58,7.96)** |
| 2.77 (0.28,27.70) | 1.67 (0.12,24.01) | 1.44 (0.36,5.87) | 0.44 (0.04,5.22) | 3.33 (0.61,18.03) | 2.43 (0.36,16.35) | **5.02 (1.33,19.04)** |
| 1.01 (0.10,9.87) | 0.61 (0.04,8.58) | 0.53 (0.13,2.11) | 0.16 (0.01,1.88) | 1.21 (0.23,6.36) | 0.89 (0.14,5.76) | 1.83 (0.51,6.66) |
| 0.98 (0.09,10.33) | 0.59 (0.04,8.89) | 0.51 (0.11,2.29) | 0.16 (0.01,1.95) | 1.18 (0.20,6.84) | 0.86 (0.12,6.13) | 1.78 (0.43,7.36) |
| 0.84 (0.09,8.25) | 0.50 (0.04,7.16) | 0.44 (0.11,1.77) | 0.13 (0.01,1.57) | 1.01 (0.19,5.32) | 0.74 (0.11,4.77) | 1.52 (0.41,5.60) |
| 1.18 (0.12,11.45) | 0.71 (0.05,9.97) | 0.61 (0.16,2.35) | 0.19 (0.02,2.15) | 1.41 (0.27,7.32) | 1.03 (0.17,6.11) | 2.13 (0.59,7.72) |
| lateral wedged insole | 0.60 (0.06,5.92) | 0.52 (0.07,3.83) | 0.16 (0.01,2.72) | 1.20 (0.14,10.28) | 0.88 (0.09,9.00) | 1.81 (0.28,11.86) |
| | knee brace | 0.87 (0.08,9.64) | 0.27 (0.01,6.13) | 2.00 (0.16,25.23) | 1.46 (0.10,21.53) | 3.02 (0.30,30.47) |
| | 0.11 (0.00,7.33) | exercise | 0.31 (0.04,2.43) | 2.30 (0.67,7.92) | 1.68 (0.38,7.46) | **3.48 (1.78,6.81)** |
| | | | hydrotherapy | 7.53 (0.71,80.24) | 5.50 (0.45,67.98) | **11.38 (1.36,95.37)** |
| | 0.05 (0.00,3.10) | 0.45 (0.03,6.57) | | KT | 0.73 (0.14,3.75) | 1.51 (0.53,4.28) |
| | 0.04 (0.00,3.44) | 0.36 (0.02,8.51) | | 0.80 (0.05,12.78) | ESWT | 2.07 (0.52,8.22) |
| | 0.04 (0.00,1.44) | 0.32 (0.05,2.29) | | 0.71 (0.11,4.43) | 0.89 (0.08,10.52) | placebo |

providing concentrated laser energy over a brief period, penetrates deeper into tissues, eliciting a more potent biostimulative and anti-inflammatory response [35]. Ahmad [11] et al., through a systematic review, have demonstrated the superiority of high-intensity laser therapy over low-intensity laser therapy in treating knee osteoarthritis, aligning with the findings of this study.

Exercise plays an extensive role in KOA management. Goh et al. [6] conducted a network meta-analysis to assess the efficacy and safety of various exercises for KOA. The American College of Sports Medicine categorizes exercises into muscle strengthening, aerobic activities, flexibility practices, and neuromuscular training. Substantial evidence supports the improvement of KOA symptoms, including pain, functionality, and quality of life, through exercise intervention [10,36–39]. Exercise modalities for KOA are diverse, with aerobic and mind-body exercises showing the most significant benefits for pain and function, while strengthening and flexibility/skill exercises are the next best options. Mixed exercises are deemed the least effective for knee and hip osteoarthritis [6,40]. A prior network meta-analysis conducted by Mo et al. [41] investigated the clinical efficacy of five different exercise therapies for the management of knee osteoarthritis, including aquatic exercise (AE), stationary cycling (CY), resistance training (RT), traditional exercise (TC), and yoga (YG). The findings concluded that AE (for pain relief) and YG (for alleviating joint stiffness, improving knee function, and enhancing quality of life) are the most effective interventions, followed by RT, CY, and TC. This study considers various exercise therapies as a single intervention for comparison with other treatment modalities, indicating that exercise therapy is effective in improving knee joint function and warrants broader clinical application.

Pulsed ultrasound's mechanical and thermal effects offer a potential treatment option for mild to moderate KOA, alleviating pain, enhancing mobility, accelerating tissue healing, and reducing edema and disability [42]. However, the role of

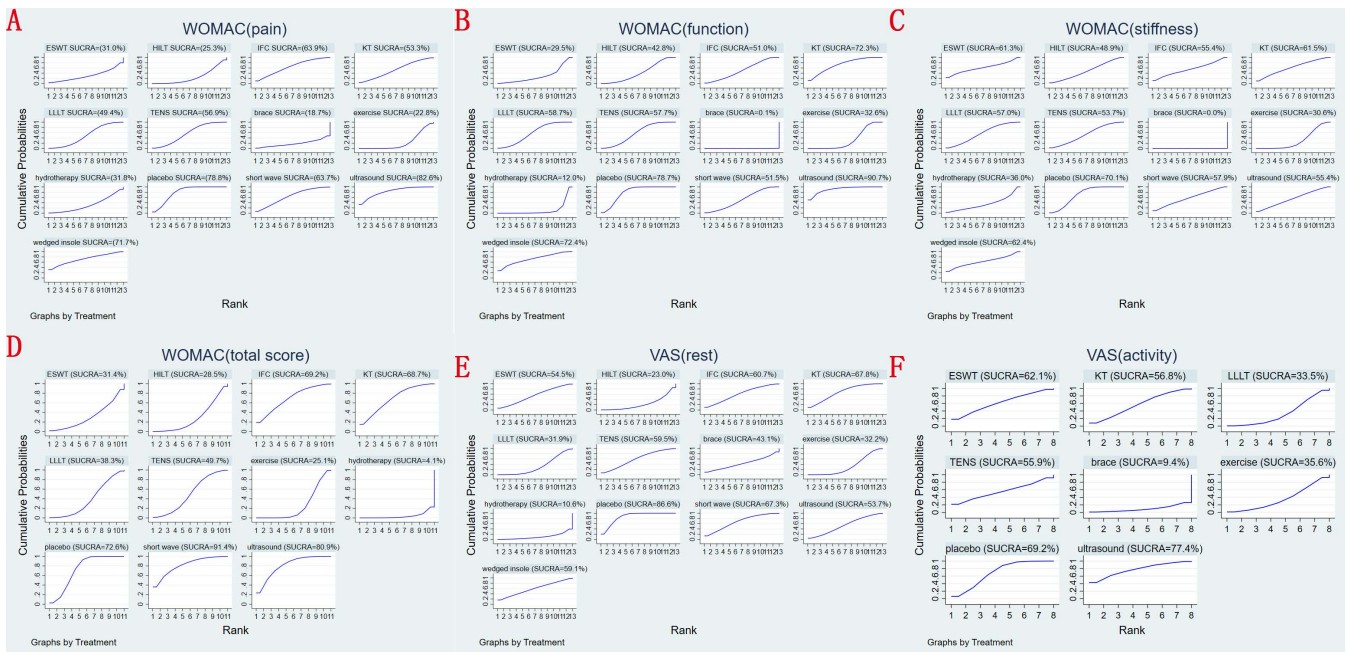

**Fig 5. Surface under the cumulative ranking (SUCRA) for (A) WOMAC pain score, (B) WOMAC function score, (C) WOMAC stiffness score, (D) total WOMAC score, (E) VAS-rest and (F) VAS-activity at last follow-up.**

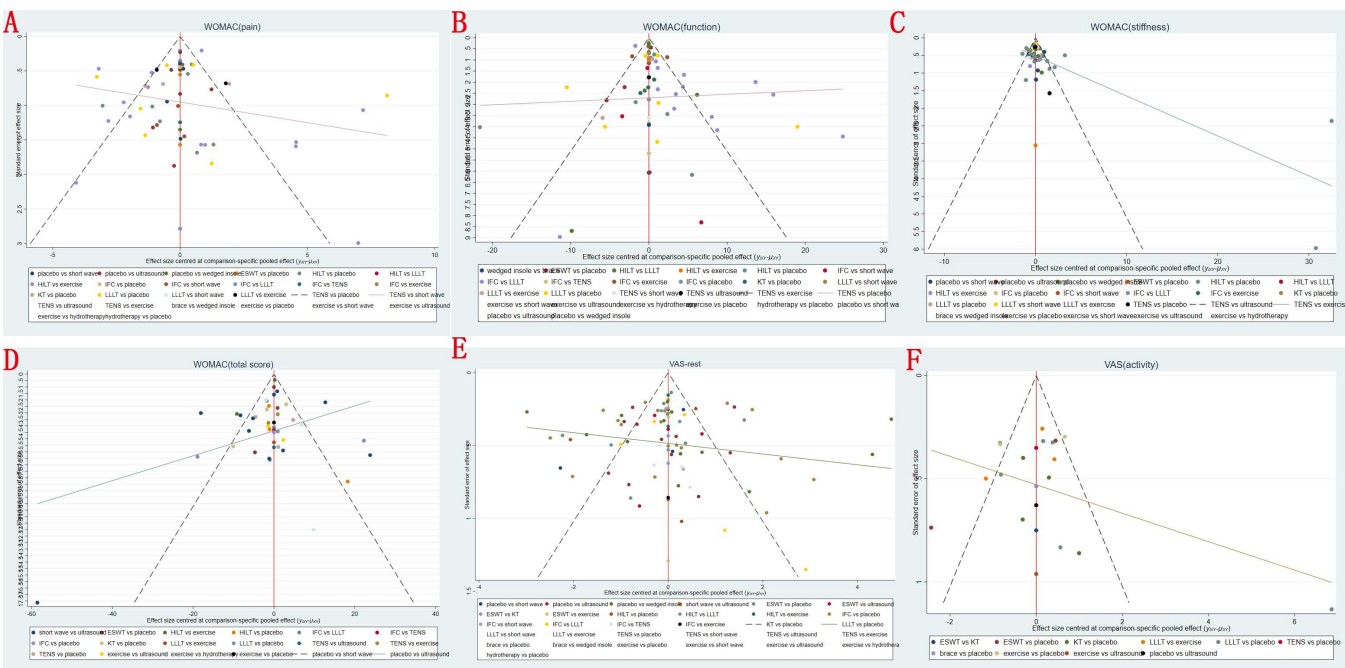

**Fig 6. Funnel plots of (A) WOMAC pain score, (B) WOMAC function score, (C) WOMAC stiffness score, (D) total WOMAC score, (E) VAS-rest and (F) VAS-activity at last follow-up.**

ultrasound pulse therapy in KOA remains contentious. Previous studies have applied unfocused continuous ultrasound at 1 or 1.5 MHz and 1-2.5 W/cm to the muscles or tendons around the knee [43,44], primarily inducing thermal effects to boost blood circulation and ease muscle and tendon spasms. Some research suggests that this thermal effect may not sufficiently address spasms due to inadequate energy absorption by the muscles, which could account for the temporary pain relief observed with pulsed ultrasound therapy in KOA [45,46]. This study's conclusions align with these findings, noting that while pulsed ultrasound is moderately effective for pain improvement, it does not significantly enhance other knee joint functions.

Wedge insoles, similar to knee orthoses, primarily aim to alleviate knee joint stress by modifying joint load. Clinically, lateral wedge insoles are predominantly utilized. However, studies indicate that they do not outperform neutral devices in pain reduction. The reduction of the knee adduction moment by only 5%-6% may be insufficient for pain alleviation. Additionally, factors like the sagittal moment and muscle co-contraction might have a more profound impact on the medial knee load, suggesting that a decrease in the adduction moment alone is not adequate to improve KOA function and pain [47–49].

When evaluating overall treatment efficacy, emphasis is placed on the WOMAC score and the resting VAS pain score, as these assessments are nearly universally conducted in the literature, thus offering a more robust framework for both direct and indirect comparative analyses. While other scale assessments are equally important, they lack the same level of comparative data, such as comparisons between the total WOMAC score and the VAS pain score during physical activity.

This study acknowledges several limitations. Firstly, variations in the duration of physical therapy across studies introduce a degree of heterogeneity in the included literature. Secondly, the majority of included studies have small sample sizes, and there are variations in gender distribution. Thirdly, despite the widespread use of WOMAC and VAS scores, there are inconsistencies in other assessment metrics. Fourthly, the methodology of the included studies, particularly regarding randomization and blinding, exhibits certain inadequacies. Lastly, the inclusion of English-language literature only may introduce language bias. Future research should prioritize evaluating the clinical efficacy of combined therapeutic approaches for KOA, as well as their cost-effectiveness and associated health-related expenses.

## 5 Conclusion

In conclusion, the findings suggest that knee brace may be the most recommended therapeutic option for the knee osteoarthritis followed by hydrotherapy and exercise.

## Supporting information

**S1 Checklist. PRISMA 2020 checklist.**
(DOCX)

**S1 File. Minimal data set.**
(XLS)

**S1 Table. Searching strategy for PubMed.**
(DOCX)

**S2 Table. Table of all data extracted.**
(DOC)

## Author contributions

**Conceptualization:** Xiao Chen, Yuan Luo.

**Data curation:** Xiao Chen, Yuan Luo.

**Formal analysis:** Xiao Chen, Yuan Luo.

**Funding acquisition:** Xiao Chen.

**Investigation:** Xiao Chen, Yuanhe Fan, Hongliang Tu, Yuan Luo.

**Methodology:** Xiao Chen, Yuan Luo.

**Project administration:** Xiao Chen, Yuanhe Fan, Hongliang Tu, Yuan Luo.

**Resources:** Xiao Chen.

**Software:** Xiao Chen.

**Supervision:** Yuanhe Fan, Hongliang Tu.

**Validation:** Xiao Chen, Yuanhe Fan, Hongliang Tu, Yuan Luo.

**Visualization:** Xiao Chen, Yuanhe Fan, Hongliang Tu, Yuan Luo.

**Writing – original draft:** Xiao Chen, Yuan Luo.

**Writing – review & editing:** Xiao Chen, Yuan Luo.

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
