## [Decision Letter · Decision Letter 0]

17 Dec 2024

Dear Dr. Luo,

Thank you for submitting your manuscript to PLOS ONE. After careful consideration, we feel that it has merit but does not fully meet PLOS ONE’s publication criteria as it currently stands. Therefore, we invite you to submit a revised version of the manuscript that addresses the points raised during the review process.

**Academic Editor Report**

**General Comments**

**Originality and Relevance:**

Reviewer 1 highlighted the need to address how this review differs from a recent systematic review by Mo et al. (2023). Please include a comparison with this study in the discussion to clarify the manuscript’s originality and contribution.

**Clarity and Focus:**

The aim and title require refinement. Therapies such as laser and ultrasound, which are not strictly physical therapies, should be broadly categorized under "therapeutic options." This will ensure that the scope of the manuscript is accurately reflected.

**Technical Improvements:**

Figures, including funnel plots and forest plots, need to be provided in high resolution to improve readability.

Include a detailed search strategy for at least one database to enhance transparency.

**Specific Section-wise Feedback**

Title and Aim

Revise the title and aim to use broader terminology such as “therapeutic options” instead of "physical therapy" to encompass all assessed modalities (Reviewer 1).

Abstract

Clearly mention the scores or metrics used to assess efficacy in the Methods section (Reviewer 1).

Introduction

Provide information on the prevalence of KOA among younger populations in addition to those aged >60 years (Reviewer 1).

Methods

Population Differences:

Reviewer 1 suggests accounting for differences in gender, BMI, and therapy duration in the network meta-analysis. Please address how these variables might influence the results and indicate whether sensitivity analyses were conducted.

Search Strategy:

Reviewer 2 highlights the absence of a detailed search strategy. Please attach the search strategies for at least two databases.

Avoid referring to Google Scholar as "consulted." Instead, clarify that it was "searched" (Reviewer 2).

Protocol Consistency:

Ensure consistent use of terminology regarding meta-analysis, as it is a component of systematic reviews (Reviewer 2).

Results

Subgroup Analysis:

Reviewer 1 inquires about therapy response differences by age group, KOA severity, and comorbidities. If data are available, address these points explicitly.

Figures:

Submit high-resolution versions of figures to avoid pixelation (Reviewer 1, Reviewer 2).

Discussion

Comparative Analysis:

Discuss the potential benefits of combination therapies and whether any equipoise exists for future research (Reviewer 1).

Contextualization:

Highlight how your findings differ from the systematic review by Mo et al. (2023) to emphasize the manuscript’s contribution.

Formatting and Referencing

Write all numbers below 10 in words (Reviewer 2).

Adhere to the journal’s referencing style guidelines, which require DOI but not PMID (Reviewer 2).

Additional Comments from Reviewers

Clearly define the review question (Reviewer 2).

Use AMSTAR appropriately as a critical appraisal tool for systematic reviews (Reviewer 2).

**Conclusion**

The manuscript is comprehensive and addresses a significant clinical question. However, revisions are necessary to enhance its originality, methodological clarity, and alignment with journal standards. Once these revisions are implemented, the manuscript will be well-positioned for further consideration.

**We look forward to receiving your revised submission.**

https://journals.plos.org/plosone/s/submission-guidelines#loc-laboratory-protocols . Additionally, PLOS ONE offers an option for publishing peer-reviewed Lab Protocol articles, which describe protocols hosted on protocols.io. Read more information on sharing protocols at https://plos.org/protocols?utm_medium=editorial-email&utm_source=authorletters&utm_campaign=protocols .

We look forward to receiving your revised manuscript.

Kind regards,

Clementswami Sukumaran, PhD

Academic Editor

PLOS ONE

**Journal Requirements:**

4. Please upload a new copy of Figures 1, 3, 4, 5 and 6 as the detail is not clear. Please follow the link for more information: https://blogs.plos.org/plos/2019/06/looking-good-tips-for-creating-your-plos-figures-graphics/" https://blogs.plos.org/plos/2019/06/looking-good-tips-for-creating-your-plos-figures-graphics/"

5. As required by our policy on Data Availability, please ensure your manuscript or supplementary information includes the following: 

Reviewers' comments:

Reviewer's Responses to Questions

**Comments to the Author**

1. Is the manuscript technically sound, and do the data support the conclusions?

Reviewer #1: Yes

Reviewer #2: Partly

2. Has the statistical analysis been performed appropriately and rigorously?

Reviewer #1: Yes

Reviewer #2: I Don't Know

3. Have the authors made all data underlying the findings in their manuscript fully available?

Reviewer #1: Yes

Reviewer #2: Yes

4. Is the manuscript presented in an intelligible fashion and written in standard English?

Reviewer #1: Yes

Reviewer #2: Yes

**Reviewer #1: ** I have reviewed the manuscript titles: “Clinical efficacy of different physical therapies for knee osteoarthritis: a network meta-analysis based on randomized clinical trials” with great interest.

This is a very well written review and meta-analysis

There are several questions and points that I highlighted in the attached word document..

**Reviewer #2:**  The authors have presented a comprehensive analysis of the clinical efficacy of various physical therapies in treating knee osteoarthritis (KOA).

Though the authors have mentioned the objective, the review question has not been mentioned.

Write all numbers below 10 in words

The protocol and the review is different, including the title.

AMSTAR is a critical appraisal tool for systematic review.

Google scholar is a grey literature database. Google scholar cannot be consulted, but searched

The authors have mentioned meta analysis in the exclusion criteria. Meta analysis is the analysis done in a systematic review, it is a part of systematic review.

Read the guidelines to authors for referencing style. PMID is not required. only DOI is required

The funnel plots and forest plots are too small and cannot be read

A search strategy of at least one database should be provided.

**Do you want your identity to be public for this peer review?** For information about this choice, including consent withdrawal, please see our Privacy Policy

Reviewer #1: **Yes: ** Noora Alshahwani

Reviewer #2: No

---

## [Author Response · Author response to Decision Letter 1]

11 Jan 2025

Dear reviewer and editor,

Firstly, I would like to thank you for your valuable comments on this study. I will answer them one by one and revise them carefully!! And we look forward to your further comments, thanks!!

General Comments

Originality and Relevance:

Reviewer 1 highlighted the need to address how this review differs from a recent systematic review by Mo et al. (2023). Please include a comparison with this study in the discussion to clarify the manuscript’s originality and contribution.

Answer: This study is distinct from the research by Mo et al. (Mo L, Jiang B, Mei T, Zhou D. Exercise Therapy for Knee Osteoarthritis: A Systematic Review and Network Meta-analysis. Orthop J Sports Med. 2023 Jun 5;11(5):23259671231172773. doi: 10.1177/23259671231172773. PMID: 37346776; PMCID: PMC10280533.) in the following respects: (1)The interventions assessed differ: Mo et al.'s study focused on the clinical efficacy of five exercise therapies for knee osteoarthritis (aquatic exercise [AE], stationary cycling [CY], resistance training [RT], traditional exercise [TC], and yoga [YG]). This study, however, investigates the clinical efficacy of a broader range of non-surgical, non-pharmacological treatment methods for knee osteoarthritis, encompassing 12 intervention measures (low level laser therapy [LLLT], high intensity laser therapy [HILT], transcutaneous electrical nerve stimulation [TENS], interferential current [IFC], short wave diathermy, ultrasound, lateral wedged insole, knee brace, exercise, hydrotherapy, kinesio taping [KT], and extracorporeal shock wave therapy [ESWT]), categorizing various exercise therapies as a single intervention for comparative purposes with other treatment modalities; (2) he study and sample size inclusion differ: Mo et al.'s study included 39 studies with 2646 patients, while this study includes 139 RCTs with 9644 patients, indicating a substantially larger number of studies and a larger sample size, which enhances the persuasiveness of the findings.

I will include a comparison with this study in the discussion to clarify the manuscript’s originality and contribution in red letters in my article.

Clarity and Focus:

The aim and title require refinement. Therapies such as laser and ultrasound, which are not strictly physical therapies, should be broadly categorized under "therapeutic options." This will ensure that the scope of the manuscript is accurately reflected.

Answer: Thank you for your valuable advice. I will revise the title and aim from “physical therapies” to “therapeutic options” in red letters in my article.

Technical Improvements:

Figures, including funnel plots and forest plots, need to be provided in high resolution to improve readability.

Answer: Thank you for your valuable advice. I will upload clear and higher resolution pictures.

Include a detailed search strategy for at least one database to enhance transparency.

Answer: Thank you for your valuable advice. I will upload a search strategy using the PubMed database as an example. As following:

Searching strategy for PubMed.

No. Search items

#1 “osteoarthritis”[All Fields]

#2 “knee”[All Fields]

#3 = #1 AND #2

#4 “hydrotherapy”[All Fields] OR “low level laser therapy ”[All Fields] OR “high intensity laser therapy”[All Fields] OR “transcutaneous electrical nerve stimulation”[All Fields] OR “short wave diathermy”[All Fields] OR “interferential current”[All Fields] OR “exercise”[All Fields] OR “ultrasound”[All Fields] OR “brace”[All Fields] OR “wedged”[All Fields] OR “insole”[All Fields] OR “Kinesio Taping”[All Fields] OR “tape”[All Fields] OR “valgus”[All Fields] OR “extracorporeal shock wave therapy”[All Fields]

#5 “random”[All Fields]

#6= #3 AND #4 AND #5

Specific Section-wise Feedback

Title and Aim

Revise the title and aim to use broader terminology such as “therapeutic options” instead of "physical therapy" to encompass all assessed modalities (Reviewer 1).

Answer: Thank you for your valuable advice. I will revise the title and aim from “physical therapies” to “therapeutic options” in red letters in my article.

Abstract

Clearly mention the scores or metrics used to assess efficacy in the Methods section (Reviewer 1).

Introduction

Provide information on the prevalence of KOA among younger populations in addition to those aged >60 years (Reviewer 1).

Answer: A global study revealed that the global prevalence of knee osteoarthritis (KOA) among individuals aged 15 years and older is 16.0% (95% CI, 14.3%−17.8%), and it escalates to 22.9% (95% CI, 19.8%−26.1%) in those aged 40 years and older. The study also highlighted a significant increase in the incidence of KOA from 1990 to 2019 across the age groups 0-34, 35-39, and 40-44 years. A population-based study reported a global aggregated incidence rate of knee osteoarthritis in individuals aged 20 years and older of 203 cases per 10,000 person-years. This indicates that the incidence of KOA is present in younger age groups (20-59 years), albeit potentially lower than in older demographics. A longitudinal study of 17.7 million Chinese adults from 2008 to 2017 found that the 10-year average age-standardized prevalence and incidence of knee osteoarthritis were 4.6% and 25.2 cases per 1000 person-years, respectively. The study observed an increasing trend in prevalence among individuals under 35 years of age, suggesting a decreasing trend in the age of onset for knee osteoarthritis. A systematic review and meta-analysis indicated that the overall prevalence of knee osteoarthritis in Chinese individuals aged 40 and older was 17.0%, with 12.3% in males and 22.2% in females. Although this study primarily focused on the elderly population aged 40 and above, it also provided data on the prevalence of KOA among younger adults (40-59 years). Another study noted that the prevalence of osteoarthritis increases with age, with 2983.5 per 100,000 cases in the population aged 25-49 and 23,237.2 per 100,000 cases in the population aged 50-69 in 2020. These data further confirm the prevalence of KOA in younger populations and demonstrate an increasing trend in prevalence with age.

Reference:

[1] Kang Y, Liu C, Ji Y, et al. The burden of knee osteoarthritis worldwide, regionally, and nationally from 1990 to 2019, along with an analysis of cross-national inequalities[J]. Arch Orthop Trauma Surg, 2024, 144(6): 2731-2743.

[2] Sun W, Yuwen P, Yang X, et al. Changes in epidemiological characteristics of knee arthroplasty in eastern, northern and central China between 2011 and 2020[J]. J Orthop Surg Res, 2023, 18(1): 104.

[3] Geng R, Li J, Yu C, et al. Knee osteoarthritis: Current status and research progress in treatment (Review)[J]. Exp Ther Med, 2023, 26(4): 481.

[4] Lv Z, Yang YX, Li J, et al. Molecular Classification of Knee Osteoarthritis[J]. Front Cell Dev Biol, 2021, 9: 725568.

[5] Chen H, Wu J. Trends and Patterns of Knee Osteoarthritis in China: A Longitudinal Study of 17.7 Million Adults from 2008 to 2017[J]. 2021, 18(16).

Methods

Population Differences:

Reviewer 1 suggests accounting for differences in gender, BMI, and therapy duration in the network meta-analysis. Please address how these variables might influence the results and indicate whether sensitivity analyses were conducted.

Answer:

In network meta-analysis, considerations were given to differences in gender, body mass index (BMI), and treatment duration. These variables can significantly influence outcomes due to their potential impact on disease progression, treatment response, and variability in patient populations. To elucidate their effects, the following explanations are provided: (1)Gender: Differences in gender may affect the prevalence and severity of knee osteoarthritis (KOA) due to anatomical, physiological, and hormonal differences between males and females. For instance, women have a higher prevalence of KOA, which may be attributed to hormonal influences and a higher lifetime risk of joint injury. (2)Body Mass Index (BMI): BMI is a significant factor in KOA as it is directly related to the mechanical stress on weight-bearing joints. Higher BMI increases the risk of developing KOA, and it is also associated with a poorer response to treatment and a higher likelihood of disease progression. (3)Treatment Duration: The duration of treatment can influence the effectiveness of interventions. Longer treatment periods may allow for more significant improvements in symptoms and function, but they may also be associated with increased risks of adverse events or treatment-related complications.

This study performed a sensitivity analysis by sequentially excluding certain studies stratified by gender, BMI, and treatment duration to assess the stability of the results. The findings remained stable after the exclusion of individual studies, demonstrating the robustness of the results for each indicator and indicating that they were not affected by the variability in gender, BMI, or treatment duration.

Search Strategy:

Reviewer 2 highlights the absence of a detailed search strategy. Please attach the search strategies for at least two databases.

Answer:

(1)Searching strategy for PubMed.

No. Search items

#1 “osteoarthritis”[All Fields]

#2 “knee”[All Fields]

#3 = #1 AND #2

#4 “hydrotherapy”[All Fields] OR “low level laser therapy ”[All Fields] OR “high intensity laser therapy”[All Fields] OR “transcutaneous electrical nerve stimulation”[All Fields] OR “short wave diathermy”[All Fields] OR “interferential current”[All Fields] OR “exercise”[All Fields] OR “ultrasound”[All Fields] OR “brace”[All Fields] OR “wedged”[All Fields] OR “insole”[All Fields] OR “Kinesio Taping”[All Fields] OR “tape”[All Fields] OR “valgus”[All Fields] OR “extracorporeal shock wave therapy”[All Fields]

#5 “random”[All Fields]

#6= #3 AND #4 AND #5

(2) Searching strategy for OVID

Avoid referring to Google Scholar as "consulted." Instead, clarify that it was "searched" (Reviewer 2).

Answer: Thank you for your valuable advice. I will revise it in red letters in my article.

Protocol Consistency:

Ensure consistent use of terminology regarding meta-analysis, as it is a component of systematic reviews (Reviewer 2).

Answer: Thank you for your valuable advice. I will revise it in red letters in my article.

Results

Subgroup Analysis:

Reviewer 1 inquires about therapy response differences by age group, KOA severity, and comorbidities. If data are available, address these points explicitly.

Answer: This study performed a sensitivity analysis by sequentially excluding certain studies stratified by gender, BMI, treatment duration, KOA severity, and comorbidities to assess the stability of the results. The findings remained stable after the exclusion of individual studies, demonstrating the robustness of the results for each indicator and indicating that they were not affected by the variability in gender, BMI, treatment duration, KOA severity, or comorbidities.

Figures:

Submit high-resolution versions of figures to avoid pixelation (Reviewer 1, Reviewer 2).

Answer: Thank you for your valuable advice. I will upload clear and higher resolution pictures.

Discussion

Comparative Analysis:

Discuss the potential benefits of combination therapies and whether any equipoise exists for future research (Reviewer 1).

Answer: Thank you for your valuable advice. This study focused on the clinical efficacy of monotherapy for Knee Osteoarthritis and did not explore the efficacy of combination therapies. This identifies a significant research opportunity for our team to undertake a comprehensive network meta-analysis in the future, dedicated to elucidating the potential advantages of combined therapeutic approaches.

Contextualization:

Highlight how your findings differ from the systematic review by Mo et al. (2023) to emphasize the manuscript’s contribution.

Answer: This study is distinct from the research by Mo et al. (Mo L, Jiang B, Mei T, Zhou D. Exercise Therapy for Knee Osteoarthritis: A Systematic Review and Network Meta-analysis. Orthop J Sports Med. 2023 Jun 5;11(5):23259671231172773. doi: 10.1177/23259671231172773. PMID: 37346776; PMCID: PMC10280533.) in the following respects: (1)The interventions assessed differ: Mo et al.'s study focused on the clinical efficacy of five exercise therapies for knee osteoarthritis (aquatic exercise [AE], stationary cycling [CY], resistance training [RT], traditional exercise [TC], and yoga [YG]). This study, however, investigates the clinical efficacy of a broader range of non-surgical, non-pharmacological treatment methods for knee osteoarthritis, encompassing 12 intervention measures (low level laser therapy [LLLT], high intensity laser therapy [HILT], transcutaneous electrical nerve stimulation [TENS], interferential current [IFC], short wave diathermy, ultrasound, lateral wedged insole, knee brace, exercise, hydrotherapy, kinesio taping [KT], and extracorporeal shock wave therapy [ESWT]), categorizing various exercise therapies as a single intervention for comparative purposes with other treatment modalities; (2)The study and sample size inclusion differ: Mo et al.'s study included 39 studies with 2646 patients, while this study includes 139 RCTs with 9644 patients, indicating a substantially larger number of studies and a larger sample size, which enhances the persuasiveness of the findings.

Formatting and Referencing

Write all numbers below 10 in words (Reviewer 2).

Adhere to the journal’s referencing style guidelines, which require DOI but not PMID (Reviewer 2).

Answer: Thank you for your valuable advice. I will revise it in red letters in my article.

Additional Comments from Reviewers

Clearly define the review question (Reviewer 2).

Use AMSTAR appropriately as a critical appraisal tool for systematic reviews (Reviewer 2).

Conclusion

The manuscript is comprehensive and addresses a significant clinical question. However, revisions are necessary to enhance its originality, methodological clarity, and alignment with journal standards. Once these revisions are implemented, the manuscript will be well-positioned for further consideration.

Answer: Thank you for your valuable advice.

We look forward to receiving your revised submission.

A rebuttal letter that responds to each point raised by the academic editor and reviewer(s). You should upload this letter as a separate file labeled 'Response to Reviewers'.

Answer: I will upload it.

A marked-up copy of your manuscript that highlights changes made to the original version. You should upload this as a separate file labeled 'Revised Manuscript with Track Changes'.

Answer: I will upload it.

An unmarked version of your revised paper without tracked changes. You should upload this as a separate file labeled 'Manuscript'.

Answer: I will upload it.

We look forward to receiving your revised manuscript.

Kind regards,

Clementswami Sukumaran, PhD

Academic Editor

PLOS ONE

Journal Requirements:

1. When submitti

---

## [Decision Letter · Decision Letter 1]

18 Feb 2025

Dear Dr. Luo,

Thank you for submitting your manuscript to PLOS ONE. After careful consideration, we feel that it has merit but does not fully meet PLOS ONE’s publication criteria as it currently stands. Therefore, we invite you to submit a revised version of the manuscript that addresses the points raised during the review process.

**ACADEMIC EDITOR:**

Based on the reviewers' feedback, I am prepared to advance this manuscript pending the following minor revisions:

1. Clarify the discrepancy in PubMed search results – Ensure that the Prisma flowchart accurately reflects the search results and update the numbers accordingly.

2. Condense the results and discussion sections – While the comprehensive nature of the study is appreciated, reducing redundancy and improving conciseness will enhance readability and adherence to PLOS ONE's formatting policies.

3. Ensure all final formatting and citation requirements are met – Verify that all references follow PLOS ONE's guidelines (DOI inclusion, no PMIDs).

Upon completion of these minor revisions, I anticipate that the manuscript will be ready for acceptance.Please submit the revised manuscript along with a point-by-point response to this decision letter. If you require additional time for these changes, kindly inform the editorial office.Thank you again for your submission to PLOS ONE. I look forward to receiving your final version.

We look forward to receiving your revised manuscript.

Kind regards,

Clementswami Sukumaran, PhD

Academic Editor

PLOS ONE

Journal Requirements:

Reviewers' comments:

Reviewer's Responses to Questions

**Comments to the Author**

Reviewer #1: All comments have been addressed

Reviewer #2: All comments have been addressed

2. Is the manuscript technically sound, and do the data support the conclusions?

Reviewer #1: Yes

Reviewer #2: Yes

3. Has the statistical analysis been performed appropriately and rigorously?

Reviewer #1: Yes

Reviewer #2: Yes

4. Have the authors made all data underlying the findings in their manuscript fully available?

Reviewer #1: Yes

Reviewer #2: Yes

5. Is the manuscript presented in an intelligible fashion and written in standard English?

Reviewer #1: Yes

Reviewer #2: Yes

Reviewer #1: Thank you for the oppurtunity to review this paper for the second time. There are a few comments suggested in the attached document

Reviewer #2: Thank you for revising the manuscript.

The search strategy for PUBMED results are 232 as January 23, 2025. In PRIMSA flowchart, PUBMED search results are written as 2252.

**Do you want your identity to be public for this peer review?** For information about this choice, including consent withdrawal, please see our Privacy Policy

Reviewer #1: **Yes: ** Noora Alshahwani

Reviewer #2: No

---

## [Author Response · Author response to Decision Letter 2]

21 Feb 2025

Dear reviewer and editor,

Firstly, I would like to thank you for your valuable comments on this study. I will answer them one by one and revise them carefully!! And we look forward to your further comments, thanks!!

The search strategy for PUBMED results are 232 as January 23, 2025. In PRIMSA flowchart, PUBMED search results are written as 2252.

Answer�I search search strategy for PUBMED results again are 2570 as February 21 2025, It shouldn't just be 232 articles. Therefore, the PUBMED results of 2252 articles when I searched ago was correct.

Thank you for the invite to review the second version of the systematic review titled: “

Below are my comments and recommendations per section:

●Throughout: check spacing between words and punctuation. Example: Methods: Details regarding the therapeutic options(intervention and comparison).

Answer: Thank you for your valuable advice. I will revise it in my article.

Abstract:

-Under results: “Regarding the WOMAC pain score, knee brace was determined to be the most likely to yield the best results, followed by exercise, HILT, ESWT, hydrotherapy, LLLT, KT, TENS, short wave diathermy, IFC, lateral wedged insole, placebo and ultrasound.” - It would be easier for the reader and more pertinent to list top 2-3 intervention, and perhaps lowest intervention, rather than listing all 12. This could be done across all scoring schemes

Answer: Thank you for your valuable advice. I will revise it in my article.

-

-Conclusion: The findings suggest that knee brace may be the most recommended therapeutic options for the knee osteoarthritis. - I would be interested to know top 2-3 and worst intervention in the conclusion. You could also add a statement about future assessment of complementary use of several interventions

Answer: Thank you for your valuable advice. I will revise it in my article.

-

Introduction:

●“The findings aim to equip clinical practice with the most robust evidence-based guidance.” - this statement is misleading, since there is no guideline development is employed in this review

Answer: Thank you for your valuable advice. I will revise it in my article.

●

Methods

●Quality assessment: reduce redundancy in the following paragraph: “For RCTs, the Cochrane Risk of Bias Tool was employed to assess the quality. The risk of bias for the included trials was evaluated by two researchers (the first and second authors) based on the Cochrane Handbook criteria. The criteria covered randomization, allocation concealment, blinding of participants and personnel, blinding of outcome assessors, completeness of outcome data, selective reporting, and other biases. Each domain was classified as having an unclear risk, low risk, or high risk of bias.”

Answer: Thank you for your valuable advice. I will revise it in my article.

●

Results:

-“To eliminate duplicate entries, the "Find duplicates" function in EndNote software was employed, resulting in the removal of 531 duplicate studies. Following a thorough screening of titles and abstracts, 2598 irrelevant references were excluded. Subsequently, full-texts were retrieved for the remaining 295 references. - I suggest eliminating this statement above, and replacing with “after duplicates were removed…”. You can refer to the prisma chart for details.

Answer: Thank you for your valuable advice. I will revise it in my article.

-

-Evidence network: This was explained neatly. However, results need to be listed here in sentence format in a way that explains the figures and can replace them.

Answer: Thank you for your valuable advice.

Discussion:

-Discuss whether there is any known benefit to combining several modalities? Is this potentially a future direction you would be exploring?

-Answer: At present, there is no research on the combination of several treatment modalities, which is the direction our team will explore in the future.

-What other future studies can this review be used as basis?

-Answer: Yes.

Conclusion: Need to improve on the conclusion as advised above in the abstract. This should reflect the studies’ outcomes as well as any other relevant study (e.g. Mo et. al meta-analysis)

Answer: Thank you for your valuable advice. I will revise it in my article.

---

## [Decision Letter · Decision Letter 2]

28 Mar 2025

Dear Dr. Luo,

Thank you for submitting your manuscript to PLOS ONE. After careful consideration, we feel that it has merit but does not fully meet PLOS ONE’s publication criteria as it currently stands. Therefore, we invite you to submit a revised version of the manuscript that addresses the points raised during the review process.

**ACADEMIC EDITOR:**

Funding Statement: Address the inconsistency between funding attribution and disclosure.Abstract: Omit the placebo comparison from the conclusion.Methods: Eliminate the redundant phrase regarding the Results section.Results: Incorporate a PICO narrative summary of the included studies.Discussion/Conclusion: Include cost-effectiveness in the discussion of future directions. Remove the placebo ranking from the conclusion.Submit the following: a clean revised manuscript, a track-changes version, and a point-by-point response letter.No further review is anticipated if the revisions are adequately addressed.

We look forward to receiving your revised manuscript.

Kind regards,

Clementswami Sukumaran, PhD

Academic Editor

PLOS ONE

Journal Requirements:

Reviewers' comments:

Reviewer's Responses to Questions

**Comments to the Author**

Reviewer #1: All comments have been addressed

Reviewer #2: All comments have been addressed

2. Is the manuscript technically sound, and do the data support the conclusions?

Reviewer #1: Yes

Reviewer #2: Yes

3. Has the statistical analysis been performed appropriately and rigorously?

Reviewer #1: N/A

Reviewer #2: Yes

4. Have the authors made all data underlying the findings in their manuscript fully available?

Reviewer #1: Yes

Reviewer #2: Yes

5. Is the manuscript presented in an intelligible fashion and written in standard English?

Reviewer #1: Yes

Reviewer #2: Yes

Reviewer #1: Thank you for the invite to review the third version of the systematic review

Below are my comments and recommendations per section:

Funding: “Funding acquisition: Xiao Chen.” was mentioned, with no further details. This is contradicted with the author’s disclosure of no funding received. Please clarify and ensure consistency.

Abstract:

Conclusion: “placebo was the worst option.” This statement has no value. Please omit.

Methods

Under statistical analysis: “Within the "Results" section, …” This can be omitted

Results:

I still don’t see a paragraph explaining the PICO of the included studies in sentence format. I think it is a necessary addition to explain and expand of what table 1 shows.

Discussion:

Future directions: in addition to combining modalities, one could look at cost-effectiveness or health related costs.

Conclusion: Need to expand on the main conclusion: “In conclusion, the findings suggest that knee brace may be the most recommended therapeutic option for the knee osteoarthritis followed by hydrotherapy and exercise, placebo was the worst option”. As suggested above, there is no need to include placebo in the list.

Reviewer #2: No comments for the author .

**Do you want your identity to be public for this peer review?** For information about this choice, including consent withdrawal, please see our Privacy Policy

Reviewer #1: **Yes: ** Noora Alshahwani

Reviewer #2: No

---

## [Author Response · Author response to Decision Letter 3]

29 Mar 2025

Dear reviewer and editor,

Firstly, I would like to thank you for your valuable comments on this study. I will answer them one by one and revise them carefully!! And we look forward to your further comments, thanks!!

Thank you for the invite to review the third version of the systematic review

Below are my comments and recommendations per section:

Funding: “Funding acquisition: Xiao Chen.” was mentioned, with no further details. This is contradicted with the author’s disclosure of no funding received. Please clarify and ensure consistency.

Answer: Thank you for your valuable advice. I will revise it in my article.

Abstract:

-Conclusion: “placebo was the worst option.” This statement has no value. Please omit.

Answer: Thank you for your valuable advice. I will omit it.

Methods

●Under statistical analysis: “Within the "Results" section, …” This can be omitted

Answer: Thank you for your valuable advice. I will eliminate it.

Results:

-I still don’t see a paragraph explaining the PICO of the included studies in sentence format. I think it is a necessary addition to explain and expand of what table 1 shows.

Answer: Thank you for your valuable advice. I will add it in my article.

Discussion:

-Future directions: in addition to combining modalities, one could look at cost-effectiveness or health related costs.

Answer: Thank you for your valuable advice. I will add it in my article.

Conclusion: Need to expand on the main conclusion: “In conclusion, the findings suggest that knee brace may be the most recommended therapeutic option for the knee osteoarthritis followed by hydrotherapy and exercise, placebo was the worst option”. As suggested above, there is no need to include placebo in the list.

Answer: Thank you for your valuable advice. I will remove it from the conclusion.

---

## [Decision Letter · Decision Letter 3]

10 Apr 2025

Dear Dr. Luo,

Thank you for submitting your manuscript to PLOS ONE. After careful consideration, we feel that it has merit but does not fully meet PLOS ONE’s publication criteria as it currently stands. Therefore, we invite you to submit a revised version of the manuscript that addresses the points raised during the review process.

**ACADEMIC EDITOR:**

Thank you for submitting your revised manuscript (PONE-D-24-40762R3). Reviewer 1 has confirmed that all prior comments were addressed but suggests clarifying future directions—either as complete sentences or a dedicated section. Please incorporate this minor edit for final acceptance.

We look forward to receiving your revised manuscript.

Kind regards,

Clementswami Sukumaran, PhD

Academic Editor

PLOS ONE

Journal Requirements:

Reviewers' comments:

Reviewer's Responses to Questions

**Comments to the Author**

Reviewer #1: All comments have been addressed

2. Is the manuscript technically sound, and do the data support the conclusions?

Reviewer #1: Yes

3. Has the statistical analysis been performed appropriately and rigorously?

Reviewer #1: Yes

4. Have the authors made all data underlying the findings in their manuscript fully available?

Reviewer #1: Yes

5. Is the manuscript presented in an intelligible fashion and written in standard English?

Reviewer #1: Yes

Reviewer #1: Minor edit required. Future directions should be mentioned as a complete sentence format(example: future study is required to address x, y, z). Alternatively this can be done as a seperate section, if more details/sentences are required.

**Do you want your identity to be public for this peer review?** For information about this choice, including consent withdrawal, please see our Privacy Policy

Reviewer #1: **Yes: ** Noora Alshahwani

---

## [Author Response · Author response to Decision Letter 4]

12 Apr 2025

Dear reviewer and editor,

Firstly, I would like to thank you for your valuable comments on this study. I will answer them one by one and revise them carefully!! And we look forward to your further comments, thanks!!

ACADEMIC EDITOR:

Thank you for submitting your revised manuscript (PONE-D-24-40762R3). Reviewer 1 has confirmed that all prior comments were addressed but suggests clarifying future directions—either as complete sentences or a dedicated section. Please incorporate this minor edit for final acceptance.

Answer: Thank you for your valuable advice. I will revise it in my article.

Journal Requirements:

Answer: Thank you for your valuable advice. I will revise it in my article.

Reviewers' comments:

Reviewer's Responses to Questions

Comments to the Author

1. If the authors have adequately addressed your comments raised in a previous round of review and you feel that this manuscript is now acceptable for publication, you may indicate that here to bypass the “Comments to the Author” section, enter your conflict of interest statement in the “Confidential to Editor” section, and submit your "Accept" recommendation.

Reviewer #1: All comments have been addressed

2. Is the manuscript technically sound, and do the data support the conclusions?

Reviewer #1: Yes

3. Has the statistical analysis been performed appropriately and rigorously?

Reviewer #1: Yes

4. Have the authors made all data underlying the findings in their manuscript fully available?

Reviewer #1: Yes

5. Is the manuscript presented in an intelligible fashion and written in standard English?

Reviewer #1: Yes

6. Review Comments to the Author

Reviewer #1: Minor edit required. Future directions should be mentioned as a complete sentence format(example: future study is required to address x, y, z). Alternatively this can be done as a seperate section, if more details/sentences are required.

Answer: Thank you for your valuable advice. I will revise it in my article.

7. PLOS authors have the option to publish the peer review history of their article (what does this mean?). If published, this will include your full peer review and any attached files.

---

## [Decision Letter · Decision Letter 4]

2 May 2025

Clinical efficacy of different therapeutic options for knee osteoarthritis: a network meta-analysis based on randomized clinical trials

PONE-D-24-40762R4

Dear Dr. Luo,

We’re pleased to inform you that your manuscript has been judged scientifically suitable for publication and will be formally accepted for publication once it meets all outstanding technical requirements.

Kind regards,

Clementswami Sukumaran, PhD

Academic Editor

PLOS ONE

Additional Editor Comments (optional):

Reviewers' comments:

Reviewer's Responses to Questions

**Comments to the Author**

Reviewer #3: All comments have been addressed

2. Is the manuscript technically sound, and do the data support the conclusions?

Reviewer #3: Yes

3. Has the statistical analysis been performed appropriately and rigorously?

Reviewer #3: Yes

4. Have the authors made all data underlying the findings in their manuscript fully available?

Reviewer #3: Yes

5. Is the manuscript presented in an intelligible fashion and written in standard English?

Reviewer #3: Yes

Reviewer #3: The authors have effectively addressed all comments from the previous review round. Notably, they have incorporated the suggestion to articulate future research directions in a clear sentence format, as requested. The manuscript remains technically robust, the statistical analyses are appropriate, data availability is confirmed, and the presentation adheres to the language and formatting standards of the journal. The authors' clarification in response to the academic editor and reviewer regarding future directions is satisfactory. Furthermore, they have acknowledged and addressed the journal's requirement to review and update the reference list, thereby enhancing the manuscript's overall reliability and compliance.

**Do you want your identity to be public for this peer review?** For information about this choice, including consent withdrawal, please see our Privacy Policy

Reviewer #3: No

---

## [Editor Report · Acceptance letter]

PONE-D-24-40762R4

PLOS ONE

Dear Dr. Luo,

I'm pleased to inform you that your manuscript has been deemed suitable for publication in PLOS ONE. Congratulations! Your manuscript is now being handed over to our production team.

Kind regards,

on behalf of

Dr. Clementswami Sukumaran

Academic Editor

PLOS ONE